# Integration of GOCI and AHI Yonsei Aerosol Optical Depth Products During the 2016 KORUS-AQ and 2018 EMeRGe Campaigns

Hyunkwang Lim[1], Sujung Go[1,2], Jhoon Kim[1], Myungje Choi[2,3], Seoyoung Lee[1], Chang-Keun Song[4], Yasuko Kasai[5]

[1]Department of Atmospheric Sciences, Yonsei University, Seoul 03722, Republic of Korea
[2]Joint Center for Earth Systems Technology, University of Maryland Baltimore County, Baltimore, MD, USA
[3]Jet Propulsion Laboratory, California Institute of Technology, Pasadena, CA, USA
[4]School of Urban and Environmental Engineering, Ulsan National Institute of Science and Technology, Ulsan 44919, Republic of Korea
[5]Natinal Institute of Information and Communications Technology, Tokyo 184-8759, Japan

*Correspondence to*: Jhoon Kim (jkim2@yonsei.ac.kr)

**Abstract.** The Yonsei AErosol Retrieval (YAER) algorithm for the Geostationary Ocean Color Imager (GOCI) retrieves aerosol optical properties only over dark surfaces, so it is important to mask pixels with bright surfaces. The Advanced Himawari Imager (AHI) is equipped with three shortwave-infrared and nine infrared channels, which is advantageous for bright-pixel masking. In addition, multiple visible and near-infrared channels provide a great advantage in aerosol property retrieval from the AHI and GOCI. By applying the YAER algorithm to 10 minute AHI or 1 hour GOCI data at 6 km × 6 km resolution, diurnal variations and aerosol transport can be observed, which has not previously been possible from low-earth-orbit satellites. This study attempted to estimate the optimal aerosol optical depth (AOD) for East Asia by data fusion, taking into account satellite retrieval uncertainty. The data fusion involved two steps: (1) analysis of error characteristics of each retrieved result with respect to the ground-based Aerosol Robotic Network (AERONET), and bias correction based on normalized difference vegetation indexes; and (2) compilation of the fused product using ensemble-mean and maximum-likelihood estimation methods (MLE). Fused results show a better statistics in terms of fraction within the expected error, correlation coefficient, root-mean-square error, median bias error than the retrieved result for each product. If the root mean square error and Gaussian center values used for MLE fusion are correct, the MLE fused products show better accuracy, but the ensemble-mean products can still be used as useful as MLE.

## 1. Introduction

Aerosols are generated by human activities and natural processes on local to global scales, and have a lifetime of several to tens of days. Aerosols affect Earth's radiative energy balance by scattering and absorption (e.g. Cho et al., 2003). High aerosol loadings are persistent in Northeast Asia, including diverse aerosol types from various sources. Interactions among aerosols, clouds, and radiation in the atmosphere cause significant uncertainties in climate-model calculations (IPCC, 2013). Datasets produced by satellites have been widely used to reduce such uncertainties (Saide et al., 2014; Pang et al., 2018), but the systems must be

accurately calibrated, verified, and consistent. Satellite data have been used extensively to
retrieve aerosol optical properties (AOPs) over broad areas, with several algorithms having
been developed. Satellites in low earth orbit (LEO), including Sun-synchronous orbit (SSO),
cover the entire Earth over one to several days, depending on instrument and orbit
characteristics. Most aerosol retrieval algorithms have been developed for LEO satellites
(Kim et al., 2007; Lyapustin et al., 2011a, b; Lee et al., 2012; Fukuda et al., 2013; Hsu et al.,
2013; Levy et al., 2013; Garay et al., 2017, 2020). LEO instruments currently onboard
satellites include the Moderate Resolution Imaging Spectrometer (MODIS), Visible Infrared
Imaging Radiometer Suite (VIIRS), Multi-angle Imaging SpectroRadiometer (MISR), and
Cloud and Aerosol Imager (CAI) (Remer et al., 2005; Lyapustin et al., 2011a, b, 2018;
Fukuda et al., 2013; Hsu et al., 2013; Levy et al., 2013; Garay et al., 2017, 2020; Jackson et
al., 2013; Lee et al., 2017).
Representative algorithms developed for MODIS data include the Dark-Target (DT; Remer
et al., 2005; Levy et al., 2013), Deep Blue (DB; Hsu et al., 2013; Sayer et al., 2014), and
Multi-Angle Implementation of Atmospheric Correction (MAIAC; Lyapustin et al., 2011a, b)
systems, which are also applied for the succeeding VIIRS (Sayer et al., 2018). In the DT
algorithm, the 2.1 μm channel is used to estimate land-surface reflectance in the visible (VIS)
region using empirical equations based on the normalized difference vegetation index
(NDVI). The DT algorithm has improved surface-reflectance modelling through
consideration of the fractional area of urbanization (Gupta et al., 2016). Ocean-surface
reflectance is estimated using the Cox and Munk method (Cox and Munk, 1954), and AOPs
over land and ocean are provided at spatial resolutions of 10 km × 10 km and 3 km × 3 km
(Remer et al., 2013), respectively. The DB algorithm has an advantage over the DT algorithm
in allowing aerosol data retrieval over bright surfaces. By using a shorter-wavelength channel,
accuracy is improved over bright surfaces such as urban and desert areas, where surface
reflectance was previously estimated by the minimum reflectance method (MRM; Herman
and Celarier 1997; Koelemeijer et al., 2003; Hsu et al., 2004). Furthermore, with the
improvement to Collection 6.1, land-surface reflectance can be estimated similarly to the DT
method, over densely vegetated regions (Sayer et al., 2019). In the case of VIIRS DB, aerosol
retrieval over the ocean is also applied by the Satellite Ocean Aerosol Retrieval (SOAR)
algorithm (Sayer et al., 2018). In the MODIS MAIAC system, surface reflectance is
estimated by considering various images based on time-series analysis, with multi-angle
observations, based on up to 16 day data, and by applying the bidirectional reflectance
distribution function (BRDF). Ocean-surface reflectance is determined using a Cox and
Munk BRDF model similar to DT and VIIRS DB (Lyapustin et al., 2011a, b, 2018). The
MISR observes Earth at nine different angles, providing a high degree of freedom for signals;
consequently, retrievals yield estimates of aerosol type and shape. As with the MAIAC,
multiple observations are used, with the estimation of land-surface reflectance involving
bidirectional reflectance factors (BRF). Zhang et al. (2016) developed an aerosol retrieval
algorithm that allows aerosol data retrieval over bright land surfaces using surface-reflectance
ratios from the VIIRS.
Aerosol retrieval algorithms for geosynchronous Earth orbit (GEO) satellites have been
developed, including the Geostationary Operational Environmental Satellite (GOES) series in
the USA (Knapp et al., 2005), Meteosat series in Europe (Bernard et al., 2011), Himawari
series in Japan (Yoon et al., 2007; Kim et al., 2008; Lim et al., 2018; Kikuchi et al., 2018;
Yoshida et al., 2018; Gupta et al., 2019), and the Geostationary Korea Multi-Purpose Satellite
(GEO-KOMPSAT, GK) series in South Korea (Kim et al., 2014, 2016; Choi et al., 2016,
2018; Kim et al., 2020). However, previously launched geostationary meteorological
satellites had only a single, broadband VIS channel, with which it is difficult to retrieve
AOPs other than aerosol optical depth (AOD) (Wang et al., 2003; Knapp et al., 2005; Kim et
al., 2008, 2014, 2016; Bernard et al., 2011). However, the Geostationary Ocean Color Imager
(GOCI) onboard the GK-1 satellite, also known as the  Communication, Ocean, and
Meteorological Satellite (COMS), has six VIS and two near-infrared (NIR) channels, which
is advantageous for retrieving AOPs (Lee et al., 2010; Choi et al., 2016, 2018; Kim et al.,
2017). Next-generation meteorological GEO satellite instruments, including the Advanced
Himawari Imager (AHI), Advanced Baseline Imager (ABI), and Advanced Meteorological
Imager (AMI), have three to four VIS and NIR channels, which enable aerosol property
retrieval with high accuracy (Lim et al., 2016, 2018; Kikuchi et al., 2018; Yoshida et al.,
2018; Gupta et al., 2019). Kikuchi et al. (2018) and Yoshida et al. (2018) performed aerosol
retrievals using the MRM and corrected reflectance using empirical equations. Gupta et al.
(2019) extended the MODIS DT algorithm to GEO satellites and estimated visible surface
reflectance using SWIR reflectance. Lim et al. (2018) retrieved the AOPs using both MRM
and estimated surface reflectance from short-wave IR (SWIR) data (ESR), and presented the
two merged products: an L2-AOD merged product, and a reprocessed AOD produced by
merging MRM and ESR surface reflectances. The MRM gives better accuracy over brighter
surfaces such as urban areas, while the ESR method gives better accuracy over areas of dense
vegetation (Lim et al., 2018). However, there is a critical surface reflectance at which aerosol
signals disappear, depending on the single-scattering albedo (Kim et al., 2016). Over the
ocean, both the MRM and ESR methods give high accuracy, but ESR results are robust with
the Cox and Munk model.
The MRM requires more computational time than the ESR method to estimate surface
reflectance, as it requires data for the past 30 days, and LER needs to be calculated using a
radiative transfer model. The ESR method estimates surface reflectance from the observed
TOA reflectance at 1.6 μm wavelength using empirical equations including the NDVI. The
advantage of MRM is that stable surface reflectance values can be obtained regardless of
surface type. However, due to the influence of background aerosol optical depth (BAOD),
surface reflectance tends to be overestimated, with satellite-derived AOD data thus being
underestimated (Kim et al., 2014). On the other hand, the ESR method uses TOA reflectance
at 1.6 μm wavelength to detect surface signals, which is less sensitive to fine particles and
BAOD. However, when aerosols such as yellow dust with coarse particles are transported
from the Taklamakan and Gobi deserts, the BAOD effect also applies to the ESR method.
The ESR method is also more likely to be affected by snow surfaces than the MRM, as snow
reduces reflectivity around the 1.6 μm wavelength (Negi and Kokhanovsky, 2011). The ESR
method also has the disadvantage of giving noisy results over bright surfaces such as desert.
However, its fast surface-reflectance estimation enables near-real-time retrieval based on the
AHI YAER algorithm.
Algorithms developed to date for LEO and GEO satellites have both advantages and
disadvantages, depending on algorithm characteristics. Therefore, the MODIS team provides
combined DT and DB AOD products (Levy et al., 2013; Sayer et al., 2014). In addition,
several studies of the fusion of L2 products have been conducted (Levy et al., 2013; Sayer et
al., 2014; Wei et al., 2019), with Bilal et al. (2017) obtaining reliable results from merged DT
and DB products, as indicated by the NDVI in East Asia, and also robust products by simply
averaging DT and DB without consideration of the NDVI.
AOP data fusion in East Asia may also be achieved using aerosol products of AMI, GOCI-2,
and the geostationary environment monitoring spectrometer (GEMS) onboard the GK-2A and
2B satellites launched by South Korea in 2018 and 2020, respectively, with accuracy over
bright surfaces being improved by the GEMS aerosol product. It is also possible to obtain

accurate AOPs, such as single-scattering albedo, aerosol loading height, and fine-mode fraction, which have been difficult to obtain by fusion of L2 data and/or surface reflectance data. If the trace-gas dataset retrieved from GEMS is used, it is possible to improve the aerosol type, with the retrieval of high-quality AOD data (Go et al., 2020).

   Several studies have considered AOD data fusion, for which methods can be broadly classified into two types. First, the fusion of more than one AOD product may involve optimal interpolation (Xue et al., 2012), linear or second-order polynomial functions (Mélin et al., 2007), arithmetic or weighted means (Gupta et al., 2008), or maximum-likelihood estimates (MLE) (Nirala, 2008; Xu et al., 2015; Xie et al., 2018). Second, in the absence of satellite-derived AOD products for the day of fusion, the geostatistical fusion method, universal kriging method (Chatterjee et al., 2010; Li et al., 2014), geostatistical inverse modelling (Wang et al., 2013), or spatial statistical data fusion (Nguyen et al., 2012) may be applied. These have the advantage that AOD can be estimated by integrating the spatial autocorrelation of AOD data even for pixels missing from the AOD products, although there is a disadvantage in not considering temporal correlations. The Bayesian maximum entropy (BME) method, taking into account temporal autocorrelation, has also been developed (Tang et al., 2016). BME methodology can estimate gap-filling pixels that are difficult to retrieve due to clouds, but with somewhat reduced accuracy. Gap filled AOD using the BME method, and satellite-derived AOD discontinuity arises from insufficient temporal sampling being available with the use of LEO satellites, resulting in a low fusion synergy. Previous studies mentioned above include data fusion based on Kriging, reproduction of spectral AOD, and BME method. Most of them focus on gap filling and rebuild AOD in areas not observed by MISR, MODIS, and SeaWiFS, and so on (Wang et al., 2013; Tang et al., 2016). However in this study, we focused on optimized AOD products with improved accuracy at the retrieved pixels by ensemble-mean and MLE fusion. We compared these two products, one very simple one and the other with more elaborated processes. As previous AOD fusion studies improved the retrieved results mainly based on MLE or NDVI-based fusion studies (Bilal et al., 2017; Levy et al., 2013; Wei et al., 2019; Go et al., 2020), we tried to further improve them with efficient approach to save computation time considering the nature of satellite data file size and user's near-real-time demand for data assimilation.
In this study, the GEO satellite dataset was used to resolve the temporal sampling issue for data fusion, while maintaining the spatio-temporal resolution retrieved from GEO satellites. We also attempted to estimate fused AOD products at 550nm with higher accuracy in East Asia. The ensemble-mean and MLE methods were applied. Section 2 describes the two algorithms used in this study for AHI and GOCI. Section 3 mentions methods of fusion and systematic bias correction, and section 4 performs validation of the fused products with the Aerosol Robotic Network (AERONET) instruments during two field campaigns: the Korea– United States Air Quality Study (KORUS-AQ) and the Effect of Megacities on the Transport and Transformation of Pollutants on Regional and Global Scales Study (EMeRGe).

## 2. Descriptions of AHI, GOCI, the YAER algorithm

### 2.1 AHI aerosol algorithm

   The Himawari-8 and -9 satellites were launched by the Japanese Meteorological Agency (JMA) on 7 October 2014 and 2 November 2016, respectively. The AHI onboard these satellites has 16 channels covering wavelengths of 0.47–13.3 μm and performs full-disk and Japan-area observations every 10 and 2.5 min, respectively, from GEO at 140.7° E longitude (Bessho et al., 2016). Visible and NIR observations are also performed at high spatial

resolutions of 0.5–1.0 km, with SWIR to IR at 2 km, which have advantages in aerosol
property retrieval and cloud masking.
Lim et al. (2018) developed the AHI Yonsei aerosol retrieval (YAER) algorithm and
provided two retrieval results with 6 km × 6 km resolution based on MRM and ESR using
SWIR data. Aerosol property retrieval using VIS channels requires accurate surface
reflectance, for which MRM and ESR are useful, with the main difference between the two
lying in the surface-reflectance estimation method.
The MRM applies the minimum-reflectance technique over both land and ocean (Lim et al.,
2018), with surface reflectance being estimated by finding the minimum reflectance in each
pixel over the past 30 day window, giving the Lambertian equivalent reflectance (LER; Kim
et al., 2016; Lim et al., 2018). This method takes the bidirectional characteristics of surface
reflectance into consideration by obtaining surface reflectance at each observation time over
the 30-day search window. However, the method assumes that there is more than one clear
day during the search window and that surface reflectance does not change; otherwise, it is
affected by clouds and/or the BAOD (Kim et al., 2014; Kim et al., 2021).
According to the ESR method, land-surface reflectance in the VIS region is constructed
from the Top of Atmosphere (TOA) reflectance at 1.6 μm wavelength, based on the NDVI
for SWIR and the fraction of urbanization and cropland (Levy et al 2013; Gupta et al., 2016;
Zhong et al., 2016; Lim et al., 2018). Ocean-surface reflectance is estimated from the Cox
and Munk BRDF model (Cox and Munk, 1954). Chlorophyll-a concentrations are considered
in addition to Chlorophyll-a concentration data
(https://www.eorc.jaxa.jp/ptree/userguide.html) from Japan Aerospace Exploration Agency
(JAXA) (Murakami et al., 2016) and interpolated for the 10-min AHI intervals. For
unretrieved pixels, the less contaminated chlorophyll-a concentration value of 0.02 mg m$^{-3}$ is
used. Details of the methodology can be found in Lim et al. (2018).
**2.2 GOCI aerosol algorithm**
GOCI is an ocean color imager in GEO launched onboard COMS in 2010 and observes the
East Asia region at an hourly interval with 500 m× 500 m resolution (Choi et al., 2012). It has
eight bands in the VIS and NIR regions, which is advantageous for aerosol retrieval. Two
versions of GOCI Yonsei aerosol algorithms have been developed, referred to as V1 and V2
(Lee et al., 2010; Choi et al., 2016, 2018). In the case of V1, surface reflectance is estimated
by the MRM using LER for the past 30 days over land, and the Cox and Munk BRDF model
over oceans. In V2, ocean-surface reflectance is estimated by the same method, but land-
surface reflectance is improved by using an accumulated long-term database. To minimize
the impact of BAOD (the weakness of the MRM), a monthly surface-reflectance database
was constructed using all of the LERs over the past five years, but it cannot reflect
unexpected changes in surface conditions. However, a well-established climatological
database allows aerosol property retrieval in near-real-time with reasonable accuracy.

**3. Data fusion methods**
Satellite-derived AODs have different error characteristics depending on NDVI, scattering
angle, and so on (Choi et al., 2016, 2018; Lim et al., 2018). Over oceans, ESR AODs are
more accurate than MRM AODs. However, the accuracy of GOCI AODs was dependent on
the NDVI values, which represent surface condition in terms of vegetation. V1 has a negative
bias and V2 has a mostly a positive bias (Choi et al., 2018). In this study, we developed
optimal AOD products at 550 nm in East Asia by fusing four individual retrievals, i.e. two
AHI aerosol products from the MRM and ESR methods, and two GOCI products from V1
and V2.

## 3.1 Spatio-temporal matching

The AHI and GOCI have different spatial pixel locations and temporal resolutions, so it is
necessary to match their spatio-temporal resolutions before data fusion. GOCI and AHI
AODs have the same spatial resolution of 6 km × 6 km, but the two satellites are located at
128.2° E and 140.7° E, respectively, at the equator. Spatial pixel matching is therefore
required. However, satellite-derived AOD represents total-column extinction, so AOD
retrieved by the two sensors is not significantly affected by satellite position. To merge the
different satellite spatial pixel coverages, the GOCI pixel was re-gridded to match AHI pixels
for full-disk observation, with up to 4 GOCI AOD pixels being used with average values
considered representative of pixel values. If more than half of the AHI AOD pixels did not
exist out of the maximum 6 AHI data per hour, it is regarded as cloud contaminated pixels
and an additional cloud removal process is performed. This process applies to both the MRM
and ESR method, to remove the AHI's additional cloud-contaminated pixels in products of
both GOCI V1 and V2, which have a disadvantage in cloud masking due to their lack of IR
channels. When three or more pixels were available for generating AHI data at 1 hour
intervals, hourly AOD values were estimated as the medians of pixel values.

## 3.2 Ensemble-mean method

Here, AMR represents AHI MRM AOD, AES represents AHI ESR AOD, GV1 represents
GOCI V1 AOD, and GV2 represents GOCI V2 AOD. We performed data fusion using AMR,
AES, GV1, and GV2 data within 1 hour intervals for which additional-cloud masking was
performed. The ensemble-mean is the mean of the ensemble member over a specific time.
The ensemble members are AMR, AES, GV1, and GV2 based on two satellite instruments
and two different surface-estimation methodologies. Table 1 provides the satellite-derived
AOD used for ensemble-mean and MLE fusion.
Fusion was performed only when a pixel of an ensemble member was used for all fusions.
Fusion 1 (F1) included the two AHI products of AMR and AES, and two GOCI products of
GV1 and GV2. Fusion 2 (F2) involved the calculation of the YAER algorithm by the fusion
of AES and GV2, both of which have the advantage of producing data in near-real-time.
Fusion 3 (F3) merged AMR and AES to estimate AOD over a wide area, and Fusion 4 (F4)
involved a comparison with F1 to determine how accuracy varied with decreasing number of
ensemble members, as summarized in Table 1.

## 3.3 MLE method

Similarly, FM1, FM2, and FM3 is the result of MLE fusion corresponding to F1, F2, and F3
as in ensemble mean, respectively (see Table 1).
The MLE method provides a means of weighting and averaging based on errors evaluated
with AERONET ground-based measurements (Nirala, 2008; Xu et al., 2015; Xie et al., 2018).
This method employs the following equations:

$$\tau_i^{MLE} = \sum_{k=1}^{N} \frac{R_{i,k}^{-2}}{\sum_{k=1}^{N} R_{i,k}^{-2}} \tau_{i,k} \tag{1}$$

$$R_{i,k} = \sqrt{\frac{\sum_{i=1}^{M}(s_{i,k} - g_i)^2}{M}} \tag{2}$$

where $\tau_i^{MLE}$ represents the fused AOD; $\tau_{i,k}$ represents the mean AOD at grid point $i$ from the
satellite-derived AOD product $k$, where $k$ is the index for different satellite-derived AOD
products for fusion; $R_{i,k}$ represents the root-mean-square error (RMSE) at grid point $i$ for the
satellite-derived AOD product $k$; $N$ is the number of all AOD data; $g_i$ represents the mean of
ground-based AOD at grid point $i$ from the AERONET (collocated temporal mean); $s_{i,k}$
represents the mean of satellite derived AOD products ($k$) at grid points of the AERONET
(collocated spatial mean); and $M$ is the number of pairs of $s_{i,k}$ and $g_i$.
279       For RMSE estimation, bias correction, validation, and error estimation (details in Sec.5),
AERONET Version 3 Level 2.0 aerosol products were used for ground truth (Giles et al.,
2019; Smirnov et al., 2000; Holben et al., 2001). RMSE and bias correction value for each
satellite product (details in Sec.3.4) required for MLE fusion were calculated through
comparison with AERONET from Apr. 2018 to Mar. 2019 excluding EMeRGe period. The
number of AERONET sites used for validation and error estimation in this study, was 35
during the KORUS-AQ campaign, and 22 during the EMeRGe campaign, for AHI and GOCI
products.
287       Satellite observation can cover wide areas, but the ground observation instrument cannot
cover all satellite observed areas. Therefore, a RMSE model was constructed for AOD, time,
and NDVI through comparative validation with AERONET observation as shown in Figure 1.
For MLE over wide areas without ground measurements, the calculated RMSE from AOD,
time, and NDVI bins was applied for every satellite pixel. We excluded points that AOD
differences with respect to AERONET data (dAOD) were > 2 standard deviations (SD) to
remove outliers and to consider only the more stable RMSE values. According to Figure 1, if
the AOD is less than 0.5, RMSE is about 0.1 with respect to all NDVI bins, but if the AOD is
greater than 0.5, the overall RMSE value becomes large. All products excluding AES show
large variations for high NDVI and high AOD bin as shown as the red square in Figure 1,
especially for 02 UTC and 05 UTC of two GOCI products and 00 UTC in AMR product.
This is because the two GOCI products and AMR are relatively less accurate for densely
vegetated areas, along with sampling issues.

## 3.4 Bias correction

301       AOD follows a log-normal distribution (Sayer and Knobelspiesse, 2019), but dAOD for
each satellite product follow a Gaussian distribution. The quantile–quantile (Q-Q) plot is a
graphical statistical technique that compares two probability distributions with each other.
The x-axis represents the quantile value of the directly calculated sample, and the y-axis
represents the Z-score. Here, the Z-score is a dimensionless value that makes a statistically
Gaussian distribution and shows where each sample is located on the standard deviation. That
is, when Z-score of 1 and 2 represent 1 SD and 2 SD, respectively. In addition, as the Q-Q
plot shows a linear shape, the sample follows a Gaussian distribution.
309       Figure 2 shows dAOD divided by SD analyzed for each satellite product, for the period
from April 2018 to March 2019, excluding the EMeRGe campaign, which shows a similar
pattern to the standard Gaussian distribution. However, if the theoretical quantile values are
greater than 0.5, then the sample quantile values are smaller than the standard Gaussian
values. Also, when the theoretical quantile is less than 0.5, the opposite results are shown.
Thus, the sample quantiles are more skewed at both sides than the theoretical quantile, but the
respective satellite product follows the Gaussian distribution.
The bias center for each satellite product was calculated differently for time and NDVI bins
through Gaussian fitting in Figure 3 of the dAOD divided by SD (except for 2SD and higher),
and subtracted from respective product for correction. Data beyond 2 SD of dAOD were
excluded to prevent a change in bias trends due to AOD errors caused by cloud shadows and
cloud contamination. This process was performed before applying the MLE method, which
allows compensation for systematic bias that is difficult to obtain directly in MLE.

### 3.5 Evaluation of aerosol products during two field campaigns


The performance of the respective satellite product and fused products was analyzed in two
field campaigns: the KORUS-AQ of 1 May 2016 to 12 Jun 2016 (https://www-
air.larc.nasa.gov/missions/korus-aq/), and the EMeRGe of 12 Mar 2018 to 8 Apr 2018
(https://www.halo.dlr.de/science/missions/emerge/emerge.html). KORUS-AQ was an
international multi-organization mission to observe air quality across the Korean Peninsula
and surrounding waters, led by the US National Aeronautics and Space Administration
(NASA) and the Korean National Institute of Environmental Research (NIER) (Crawford et
al., 2021). EMeRGe aimed to investigate experimentally the patterns of atmospheric transport
and transformation of pollution plumes originating from Eurasia, tropical and subtropical
Asian megacities, and other major population centers. GEO satellite data played an important
role in these campaigns; e.g., data assimilation for chemical transport models and tracking
aerosol plumes (Saide et al., 2014, 2010; Pang et al., 2018).
In this study, we used satellite-derived GOCI and AHI AODs, with a spatial resolution of 6
km × 6 km, and temporal resolutions of 1 hour and 10 minutes, respectively. Spatio-temporal
correlation between satellite-derived AOD and AERONET AOD involved data averaged over
all satellite pixels within a 25 km radius of the AERONET site, and AERONET AOD
averaged over ±30 minutes from the satellite observation time. As validation metrics,
Pearson's correlation coefficient, median bias error (MBE), the fraction (%) within the
expected error of MODIS DT (EE), and Global Climate Observing System requirement for
AOD (GCOS; GCOS, 2011) were applied. The accuracy requirement of GCOS for satellite-
derived AOD at 550nm is 10% or 0.03, whichever is larger. The EE provided by the MODIS
DT algorithm (EE as $\pm 0.05 \pm 0.15 \times$ AOD; (Levy et al., 2010)) was used for consistent
comparison with previous studies.
Table 2 shows the validation metrics of the respective product during the two field
campaigns. The collocation points for validation with AERONET of two AHI and two GOCI
products were not significantly different. %EE and %GCOS of AES and AMR showed better
accuracy than GV1 and GV2 during the KORUS and the EMeRGe periods. In terms of MBE,
GV2 is 0.008 and -0.001, which shows during the KORUS-AQ and the EMeRGe periods
close to zero. Additionally, further analyzes of the respective satellite product are carried out
along with fused products in Section 5.

### 4. Results


Figure 4 (a) shows the average AOD of FM1 (MLE method with all products) during the
KORUS-AQ period, and Figure 4 (b-e) shows the respective difference of the average AOD
of AMR, AES, GV1, and GV2 with respect to FM1. FM1 was selected as the representative
fused product as FM1 used all four satellite-derived products for fusion with bias correction.
The result of the comparison with the respective satellite product (Figure 4 (b-e)) shows
different features. AMR shows a negative bias over the ocean but shows similar results to
FM1 over land, while AES shows a different tendency in northern and southern China. GV1
tends to show opposite pattern to AES, and GV2 shows positive bias over the ocean and
results in similar pattern to FM1 over the land. In the west of the Korean peninsula, AES
AOD has a positive offset compared to FM1. Although the AES algorithm considers the
fraction of urbanization, there is still a tendency to have a positive AOD offsets. The main
reason why AES results show different patterns is the different estimation process of the land
surface reflectance from that of other products.
On the other hand, in GV1, the AOD over the Manchurian region has a positive offset
compared to FM1. This is because the aerosol signal is small over bright surface, making it
difficult to retrieve aerosol properties. These features tend to be alleviated in GV2, where the
surface reflectance and cloud removal process were improved.
Figure 5 shows the same result as Figure 4 except for the EMeRGe period. The AMR and
AES AODs appeared high in northern China, which is thought to be the snow contaminated
pixel. The EMeRGe period was in March-April, when northern China is more covered by
snow compared to the KORUS-AQ period in May-June. On the other hand, for GV1 and
GV2, the effect of overestimation with snow contaminated pixel is relatively small, as their
snow masking is well performed. However, for the KORUS-AQ period, it seems that the
GV1's overestimation of AOD in northern China still remains. Since this analysis (Figure 4
and 5) is for the fusion between the three MRM results and one ESR result, the average field
difference is naturally the largest in AES which uses ESR method.
For the characteristics of the average AOD for the two campaign period, high AODs during
the KORUS-AQ period were found in eastern China, and Hokkaido as wildfires from Russia
were transported to Hokkaido (Lee et al., 2019). Meanwhile, during the EMeRGe period,
high AOD is shown over the Yellow sea as aerosols were transported from China to the
Korean peninsula through the west coast, contrary to the KORUS-AQ period. Overall, the
average AODs for the EMeRGe are less smooth than those of the KORUS-AQ period. This is
because the EMeRGe period was shorter than that of the KORUS-AQ, and the retrieval
accuracy was lower due to the bright surface.

**5. Validation, comparison, and error estimation against AERONET**
**5.1 Validation for fused AOD products with AERONET**

The spatio-temporal matching method between fused AOD and AERONET was performed
as mentioned above in Section 3.5, and the statistics indices used for verification are also the
same. Validation indices of fused products with AERONET AOD during the two campaign
periods are summarized in Table 3. During the KORUS-AQ, fused AODs have better
accuracy of than respective satellite product in terms of %EE and %GCOS. The %EE
and %GCOS of AES, which showed the best accuracy among the respective product, are 63.5%
and 43.6%, which are poor than the worst accuracy of the fused AOD. All RMSE has been
improved except for FM2. The RMSE of FM2 is higher than RMSE of respective satellite
product by 0.001. Although all MBEs show different patterns, the deviation of the fused
products tends to be smaller. GV2 and F2 show MBE of 0.008, close to zero.
Next, %EE for the EMeRGe period exceeded 60.0, with AMR having the best accuracy of
69.4.  Likewise, %GCOS was also the highest with 52.4, which showed better accuracy than
the fused product. In terms of MBE, GV2 was the best, with -0.001. The fused products did
not have the best statistical values, but they show overall better statistical values.
Figure 6 shows the %GCOS for the respective satellite product and fused products at each
validation site during each campaign. In Figure 6(a), for the KORUS period, F1 and FM1
show the highest % GCOS at 20 sites out of 35. Other than the fused result, AES shows the
highest %GCOS at 13 sites, which are mostly dense vegetation-area and coastal sites.  On the
other hand, during EMeRGe period, the %GCOS of fused products was highest at 7 sites out
of 22, while respective satellite product showed at the rest of the sites in similar proportions.

**5.2 Error estimation**
Differences between satellite products and AERONET, dAOD values were analyzed in
terms of NDVI and observation times (Figure 7). Figure 7 (a) and (d) shows the respective
satellite product, Figure 7 (b) and (e) the ensemble-mean product, and Figure 7 (c) and (f) the
MLE fusion results, with each filled circle representing the mean of 500 and 400 collocated
data points sorted in terms of NDVI for the KORUS-AQ and the EMeRGe campaigns,
respectively. Figure 7 (a) shows different biases for each satellite product, with AMR and
GV1 being negative, AES and GV2 being positive. The errors are close to zero for both the
ensemble-mean and MLE products except for FM2 as a result of the fusion process.
When the NDVI is small, the Gaussian center for GV2 dAOD was close to zero, but when the
NDVI is large, the Gaussian center was negative as shown in Figure 3. The bias correction
effect of GV2 shows a small effect for small NDVI bins and a large effect for large NDVI
bins. In fact, the collocated dAODs of FM2 show close to zero when the NDVI bins are
greater than 0.4 (in Figure 7 (a)).
During the EMeRGe campaign (right column, Figure 7), the two AHI and two GOCI
products show negative biases, and even the ensemble-mean results have negative biases. The
ensemble-mean does not include any bias correction, meaning that the error characteristics of
each original satellite product are intact. The MLE products display improved biases in terms
of NDVI, which are close to zero because the bias was corrected for in the MLE process.
During the EMeRGe period, the collocated dAOD values at NDVI around 0.1 have a
negative value for all satellite-derived products (especially AHI products), and GV1 has a
negative value for bins where NDVI is greater than 0.2. During the EMeRGe period, the
collocated dAOD values at NDVI around 0.1 show negative values for all respective product
(especially AHI products), and dAOD for GV1 shows negative values for NDVI bins greater
than 0.2. The fused products tend to have error close to zero except for F3 and FM3. In terms
of F3, the collocated dAOD value around 0.1 of the NDVI bin has negative values for both
AMR and AES, so the collocated dAOD of F3 remain negative. Gaussian center values for
FM3, AMR and AES (in Figure 3) are close to zero for NDVI at around 0.1, so the bias
correction effect is small. This can be explained by the fact that the collocated dAOD for
NDVI at around 0.2 during the EMeRGe period is closer to zero in FM3 than in F3.
The median bias of the AOD products over the observation time was analyzed as shown in
Figure 8 where the left column represents the KORUS-AQ and the right column the EMeRGe
campaign, with filled circles representing median values, and the error bar being ±1 SD. As
in the KORUS-AQ campaign, the AMR shows a generally negative bias, as in the all-time
results, and a negative bias also exists in each time zone. In the AES, GV1, and GV2 case,
positive and negative biases appear differently according to time zones. The ±1 SD of the
respective satellite product is larger at local noon and smaller at 00 and 07 UTC when SZA is
large. Fused products as shown in Figures 8 (b) and (c), have a smaller ±1 SD, and the
collocated dAOD over the observation time is also close to zero. Meanwhile, FM2 shows the
same tendency of overestimation for the same reason as in the previous Figure 7(a).
For the EMeRGe period, the collocated dAOD values of the respective product appear
closer to zero than KORUS-AQ. Similarly, the collocated dAOD of the fused products also
show values close to zero.
The error analysis indicates that the results after fusion are more accurate than the results
obtained using individual satellite product, and fused products accuracy was slightly better
during KORUS-AQ than EMeRGe because more data points were considered. Also, the
surface was relatively dark during the KORUS-AQ period, thus reduced errors for aerosol
retrieval than during the EMeRGe period.

## 5.3 Time-series analysis of daily mean and hourly AODs

The Gangneung-Wonju National University site (Gangneung-WNU; 128.87°E, 37.77°N)
lies on the eastern side of the Korean Peninsula and it is one of the regions with low aerosol
loadings. The AOD frequency distribution generally follows a log-normal distribution, and it
is important to evaluate accuracy for low AOD values. Therefore, we evaluated whether the
fused products were improved at low AODs. A daily mean time-series and diurnal variation
comparison of different satellite AOD products against AERONET (on a logarithmic scale)
are shown in Figure 9 for the Gangneung-WNU site without high AOD events, where most
point AERONET AODs at 550 nm were < 1 during the KORUS-AQ campaign. Daily mean
time-series data from the AERONET, ensemble-mean, and MLE products are shown in
Figure 9 (a-c), where black filled circles and black error bar represent AERONET AOD and
±1 SD of one-day average AERONET AOD. Satellite-derived AODs represented in different
colors show similar variabilities.
Respective satellite product generally shows similar daily-mean AOD distribution to
AERONET AOD. AMR, GV1, GV2 using MRM technique show similar patterns, and AES
using SWIR for surface reflectance estimation shows different patterns. The daily-mean AOD
of AES is more close to AERONET. On the other hand, Figure 9 (b) and (c) representing
fused AOD show similar patterns overall, but the daily-mean AODs on 11 May show
different patterns. Here, ensemble-mean products (F1-4) are less accurate than an individual
AES product, while MLE products (FM1-3) exhibit similar diurnal variation to daily-mean
AERONET AOD. To further analyze this, the daily-mean AOD is shown in Figure 9 (d-f)
instead of the hourly AOD for 11- 14 May.
As in the previous daily-mean AOD results, Figure 9 (d) shows the hourly AES AOD
variations are close to hourly AERONET, while AMR, GV1, and GV2 tend to underestimate.
Similarly, as shown in Figure 9 (e), hourly AOD variation of the ensemble-mean products
shows overall underestimation for 11 May. All ensemble-mean products use AES as an
ensemble member, but do not sufficiently compensate for the negative biases held by AMR,
GV1, and GV2. Meanwhile, MLE fused products show similar patterns to the hourly AOD
variation of AERONET, such as AES outputs. This can be explained in two ways: the effect
of considering the weighted function based on pixel-level uncertainty (RMSE in this study)
and the bias correction effects. Figure 1 showed similar RMSE values for all observation
times when AOD ≤ 0.5. Gangneung-WNU site is one of the densely vegetated areas, but if
the AOD is less or equal to 0.5, there is little sensitivity of RMSE according to NDVI bins.
That is, regardless of the NDVI, each satellite-specific weighting function used for the MLE
fusion has a similar value for all satellite-derived products. The difference between the
ensemble-mean and the MLE fused products is due to the bias correction considered in the
MLE fusion. For example, the FM3 states that AMR has a large negative bias in the
afternoon and AES has a negative bias in the morning. With the bias correction of AES and
AMR respectively in the morning and afternoon, FM3 is calibrated in a direction to
compensate the underestimated AOD. The effect of bias correction and MLE fusion
agreement varies depending on the NDVI and AOD loading for each pixel. If bias correction
was not performed in the case on 11 May, the MLE fusion output shows very similar values
to F3.
The MLE products were implemented in a way to improve accuracy for the low AOD
region more critically than in the high AOD region by systematic bias correction. In general
surface reflectance estimated by the MRM is affected by BAOD, to result in a negative bias
in AOD. On the other hand, the AES uses TOA reflectance at 1.6 μm wavelength to estimate
surface reflectance and is therefore less affected by BAOD, and shows higher AOD than
AMR and the two GOCI AODs. Furthermore, AOD retrieval over vegetated areas is more
accurate with the ESR method. This result is consistent with previous studies of aerosol
retrieval in the VIS region (Levy et al., 2013; Gupta et al., 2019; Hsu et al., 2019).

**5.4 Accuracy evaluation for AHI products of the outside of GOCI domain**
In this section, the AMR, AES, F3, and FM3 products were evaluated at 34 sites within the
0-50°N and 70-150°E except for the GOCI domain as shown in Figures 4 and 5 (112-148°E,
24-50°N). The evaluation results are summarized in Table 4 in terms of N, R, RMSE, MBE,
and GCOS fraction. The RMSE and Gaussian center values within the GOCI domain were
used in the MLE fusion in this section (see Figures 1 and 3). Table 4 shows the %GCOS and
RMSE values with poor accuracy than the validation results for the GOCI coverage as listed
in Table 4. In addition, BME during the KORUS-AQ and the EMeRGe period was -0.098
and -0.135 for AMR, and 0.130 and -0.055 for AES, respectively, which show very poor
accuracy. This can be explained by the cloud contamination issue at sites near the equator,
including Thailand. In addition, since AMR cannot collect enough clear pixels for the
estimation of LER, which can cause errors. Furthermore, MRM does not work well over
desert areas. On the other hand, AES has issues with poor accuracy over bright pixels such as
desert and snow contaminated areas. Second, there are many areas where the coastline is
complex as in Hong Kong, and the surface elevation is uneven as in Himalayas. However,
there is a bias of -0.055 during the EMeRGe period for AES, but the %GCOS was the highest
with 34.1, which is considered significant. F3 and FM3 show similar patterns for the
KORUS-AQ and the EMeRGe period. The accuracy of F3 is better than that of FM3, because
the previously mentioned issue for the bias correction has worked incorrectly, as the RMSE
and bias correction values used were from the data in the untrained area.

**6. Summary and conclusion**
Various aerosol algorithms have been developed for two different GEO satellites, AHI and
GOCI. Retrieved AOD data have advantages and disadvantages, depending on the concept of
the algorithm and surface-reflectance estimations. In this study, four aerosol products (GV1,
GV2, AMR, and AES) were used to construct ensemble-mean and MLE products. For the
ensemble-mean, this study presented fusion products taking advantage of overlap region,
accuracy, and near-real-time processing. For MLE products, bias corrections for different
observation times and surface type were performed considering pixel-level errors, and the
synergy of fusion between GEO satellites was successfully demonstrated.
Validation with the AERONET confirmed that averaging ensemble members improved
most of statistical metrics for ensemble products, and consideration of pixel-level uncertainty
further improved the accuracy of MLE products. For optimized AOD products in East Asia,
NDVI and time-dependent errors have been reduced. The ensemble-mean and MLE fusion
results show consistent results with better accuracy.
By comparing F1 and F4, we can see the accuracy changes depending on the number of
members used in the ensemble-mean. During the KORUS-AQ period, poor accuracy of each
member for ensemble averaging made difficult to find true features. The accuracy of F4 was
higher than that of F1, which shows the effect of GV1's large bias during the KORUS-AQ
period. On the other hand, for the EMeRGe period, the difference between F1 and F4 appears
small because the respective ensemble member's accuracy was better. Both near-real-time
products, F2 and FM2, show good accuracy, similar to other fused products. Interestingly, the
accuracy of F1 was worse than that of F2, but the accuracy of FM1 was better than that of
FM2. The reason for this appears that the long-term RMSE (in Figure 1) and Gaussian center
value (in Figure 3) was a better representation for the EMeRGe than for the KORUS-AQ
period. To minimize such errors, overall results can be improved by binning the RMSE and
Gaussian center value for the bias correction with respect to month and season in addition to
NDVI and time. Naturally, if we directly use the RMSE and Gaussian center value of each
campaign, the accuracy can be improved.
In terms of %GCOS range, satellite-derived and fused products was 33-43% and 46-54%,
respectively during the KORUS-AQ, indicating that the fused products have a better or
similar statistical score along with other validation scores such as RMSE and MBE. However,
the %GCOS during the EMeRGe period shows better accuracy for AMR products with 52.4%
than for fused products with a maximum of 47.6%. In terms of other validation indices,
however, such as RMSE and MBE, the fused product results represent a better validation
score than the AMR. For low aerosol loading case where RMSE is small and similar across
different products, bias correction effect was also analyzed at the Gangneung-WNU site by
comparing F3 and FM3.
As a summary, to increase the accuracy of the fused products, it is required to have either
high accuracy of the respective satellite product, or the consistent error characteristics with
respect to different parameters such as time, NDVI, etc. If either each satellite-derived AOD
is accurate or large numbers of ensemble members are available for compensating respective
error, the ensemble-mean shall be the better fusion technique. If the error characteristic is not
random and can be expressed as a specific function, the fused product's accuracy through the
MLE fusion will be increased.
The method applied in this study could be used for AOD fusion of GEO data, such as AMI
onboard GK-2A, GOCI-2 and GEMS onboard GK-2B. Furthermore, it is possible to retrieve
AOPs other than AOD using multi-angle and multi-channel (UV, VIS, and IR) observations
with GK-2A and 2B.

**Code and data availability.**
The aerosol products data from AHI and GOCI are available on request from the
corresponding author (jkim2@yonsei.ac.kr).
**Author contributions.**

HL, SG and JK designed the experiment. HL and SG carried out the data processing. MC, SL,
and YK provided support on satellite data. HL wrote the manuscript with contributions from
co-authors. JK reviewed and edited the article. JK and CK provided support and supervision.
All authors analyzed the measurement data and prepared the article with contributions from
all co-authors.
**Competing interests.**
The authors declare that they have no conflict of interest.

**Acknowledgements**
We thank all principal investigators and their staff for establishing and maintaining the
AERONET sites used in this investigation. This subject is supported by Korea Ministry of
Environment (MOE) as "Public Technology Program based on Environmental Policy
(2017000160001)". This work was also supported by a grant from the National Institute of
Environment Research (NIER), funded by the Ministry of Environment (MOE) of the
Republic of Korea (NIER-2020-01-02-007). This research was also supported by the National
Strategic Project-Fine particle of the National Research Foundation of Korea (NRF) funded
by the Ministry of Science and ICT (MSIT), the Ministry of Environment (ME), and the
Ministry of Health and Welfare (MOHW) (NRF-2017M3D8A1092022). We thank all
members of the KORUS-AQ science team for their contributions to the field study and the
data processing (doi:10.5067/Suborbital/KORUSAQ/DATA01).

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

Table 1. Satellite dataset used for the fusion products. Four entries F1-F4, and three entries FM1-FM3 represent
ensemble-mean fusion and MLE fusion products.

| AOD type | F1 | F2 | F3 | F4 | FM1 | FM2 | FM3 |
|---|---|---|---|---|---|---|---|
| AER | o | o | o | o | o | o | o |
| AMR | o | | o | o | o | | o |
| GV1 | o | | | | o | | |
| GV2 | o | o | | o | o | o | |
| Remark | | | | Without | MLE Products[2] | | |
| | All available products | For NRT[1] | AHI only for wider area | GV1 to check missing effect | Same as F1 | Same as F2 | Same as F3 |

[1] NRT: near real time; [2] Maximum Likelihood Estimation

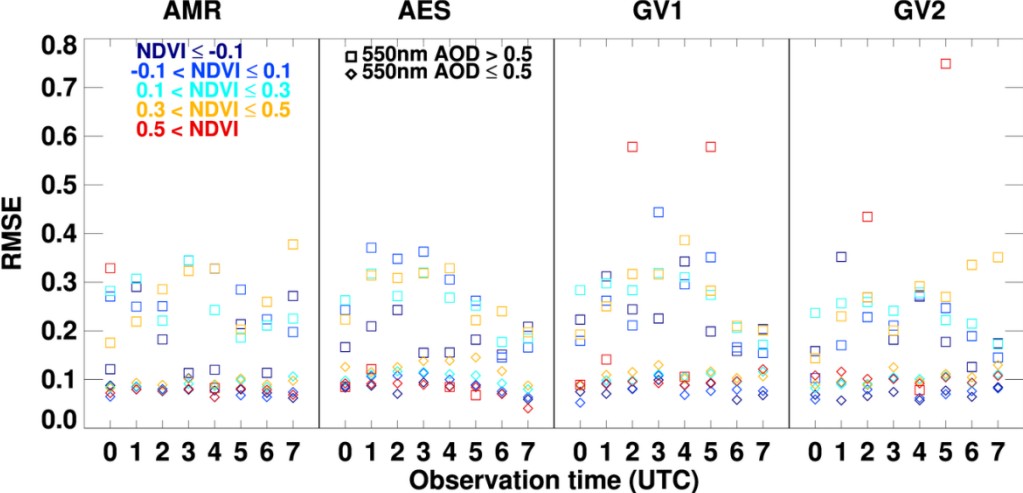

Figure 1. RMSE according to NDVI (color), observation time, and satellite AODs (square and diamond
represent AOD at 550nm greater and less equal than 0.5) during Apr. 2018 to Mar. 2019 excluding EMeRGe
campaign.


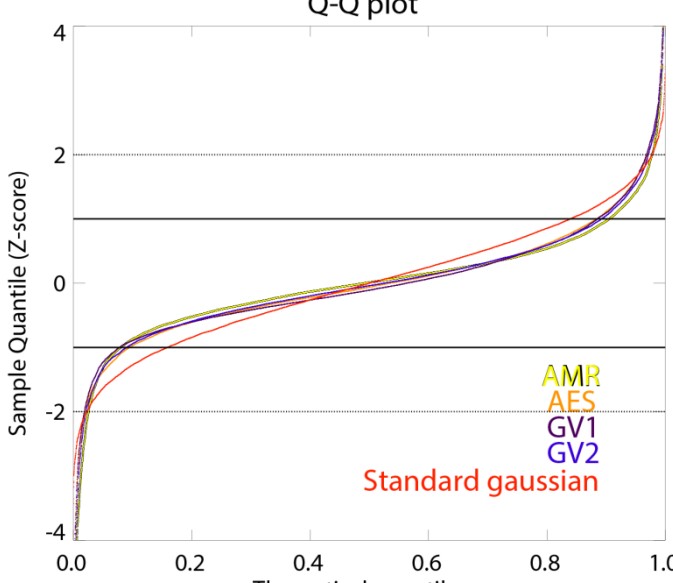

Figure 2. Q-Q plot for the difference between AERONET AOD and AMR(purple), AES(cyan), GV1(green), and
GV2(orange) AOD. The black solid line and dotted line represent 1-σ and 2-σ, respectively.

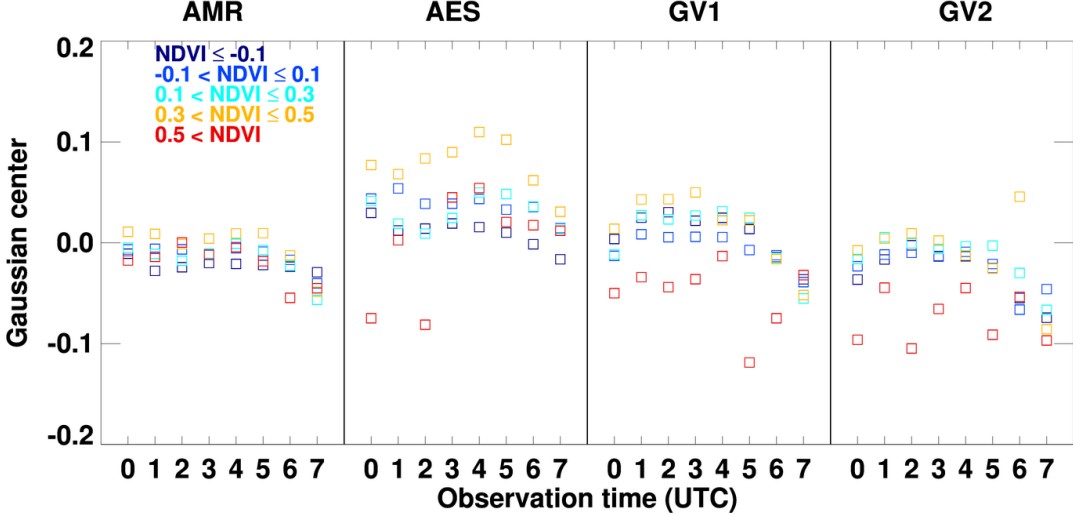

Figure 3. Systematic bias-correction values for NDVI groups and temporal bins for each satellite product from
Gaussian fitting analysis used in MLE fusion.


Table 2. Validation statistics of the respective satellite product during the KORUS-AQ and the EMeRGe
campaign.

| Product type | KORUS-AQ | | | | | EMeRGe | | | | |
|---|---|---|---|---|---|---|---|---|---|---|
| | %EE | %GCOS | RMSE | MBE | N | %EE | %GCOS | RMSE | MBE | N |
| AES | 63.5 | 43.6 | 0.145 | 0.029 | 5069 | 65.2 | 46.3 | 0.176 | -0.011 | 1884 |
| AMR | 60.6 | 39.4 | 0.150 | -0.054 | 5069 | 69.4 | 52.4 | 0.162 | -0.028 | 1884 |
| GV1 | 52.2 | 34.7 | 0.153 | -0.045 | 4843 | 63.4 | 42.7 | 0.162 | -0.035 | 1760 |
| GV2 | 50.3 | 33.8 | 0.176 | 0.008 | 4924 | 61.5 | 41.8 | 0.164 | -0.001 | 1863 |


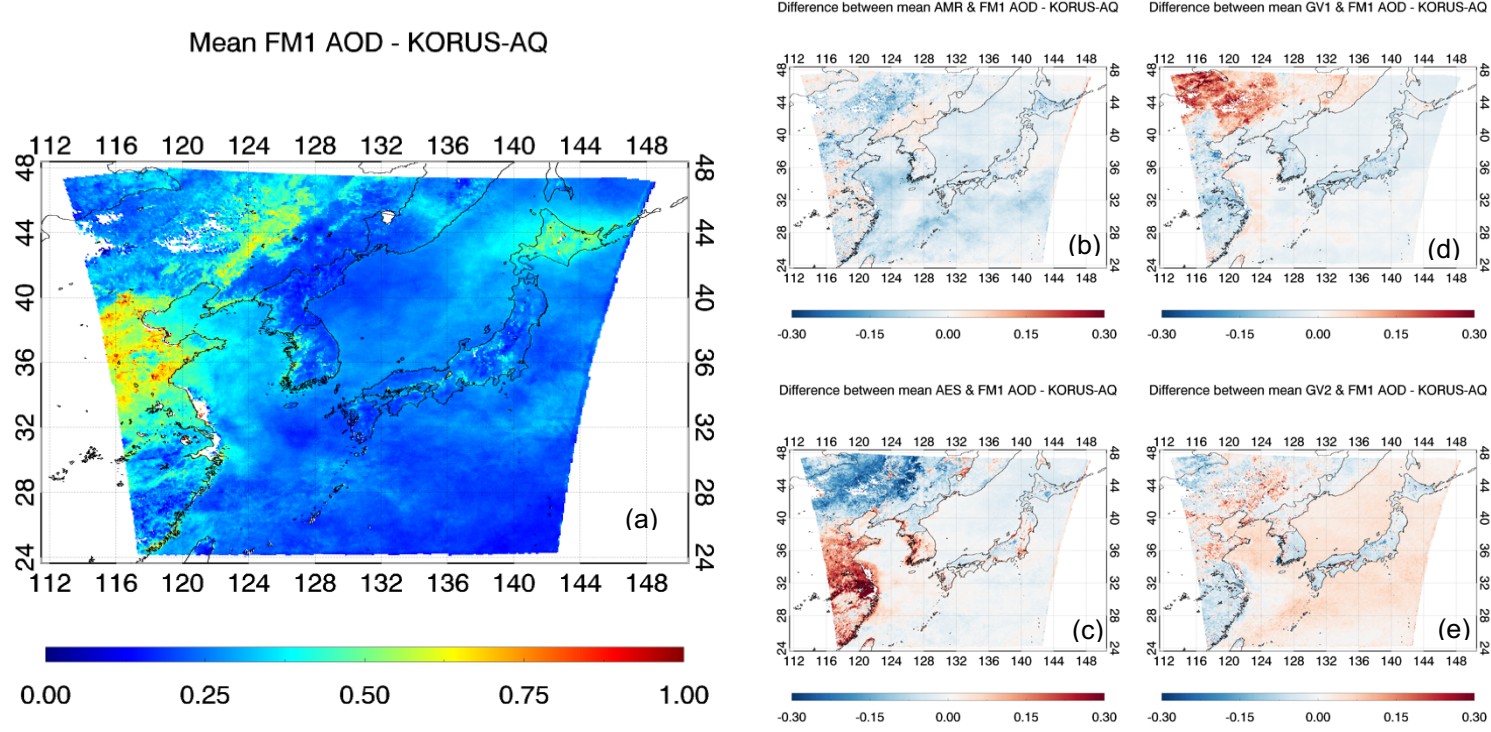

Figure 4. The average AOD of (a) FM1 (AMR, AES, GV1, and GV2) during the KORUS AQ. The difference of mean (b)AMR, (c)AES, (d)GV1, and (e)GV2 AODs with respect to mean representative (FM1) AOD. Figures generated with Interactive Data Language (IDL) version 8.8.0.

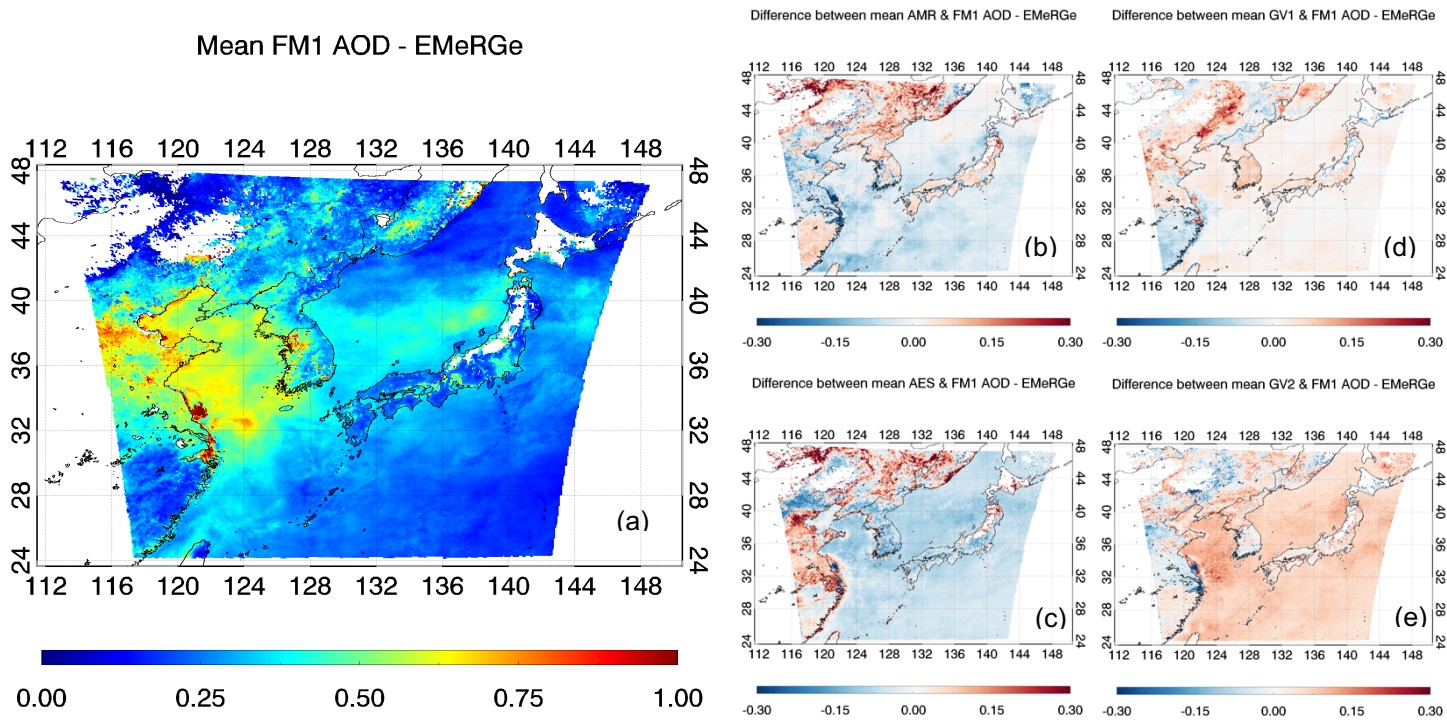

Figure 5.  Same as Figure 4, but for EMeRGe campaign.

Table 3. Validation statistics of the ensemble-mean fusion (F1-F4), and MLE fusion (FM1-FM4) AOD during two field campaigns (left: KORUS-AQ, right: EMeRGe).

| Fusion method | Product type | KORUS-AQ | | | | | EMeRGe | | | | |
|---|---|---|---|---|---|---|---|---|---|---|---|
| | | %EE | %GCOS | RMSE | MBE | N | %EE | %GCOS | RMSE | MBE | N |
| Ensemble-mean | F1 | 67.8 | 47.2 | 0.134 | -0.014 | 4806 | 66.8 | 45.4 | 0.149 | -0.012 | 1754 |
| | F2 | 72.3 | 52.7 | 0.129 | 0.008 | 4843 | 66.9 | 45.5 | 0.150 | -0.012 | 1760 |
| | F3 | 72.1 | 51.1 | 0.133 | 0.012 | 5069 | 63.2 | 44.5 | 0.175 | -0.019 | 1884 |
| | F4 | 73.3 | 51.6 | 0.128 | -0.015 | 4843 | 66.4 | 44.8 | 0.153 | -0.024 | 1760 |
| MLE | FM1 | 72.6 | 52.4 | 0.130 | -0.012 | 4806 | 69.1 | 47.6 | 0.147 | -0.008 | 1754 |
| | FM2 | 65.5 | 46.1 | 0.146 | 0.034 | 4924 | 67.3 | 46.5 | 0.152 | 0.014 | 1863 |
| | FM3 | 75.2 | 54.5 | 0.129 | -0.09 | 5069 | 62.4 | 41.8 | 0.177 | -0.027 | 1884 |

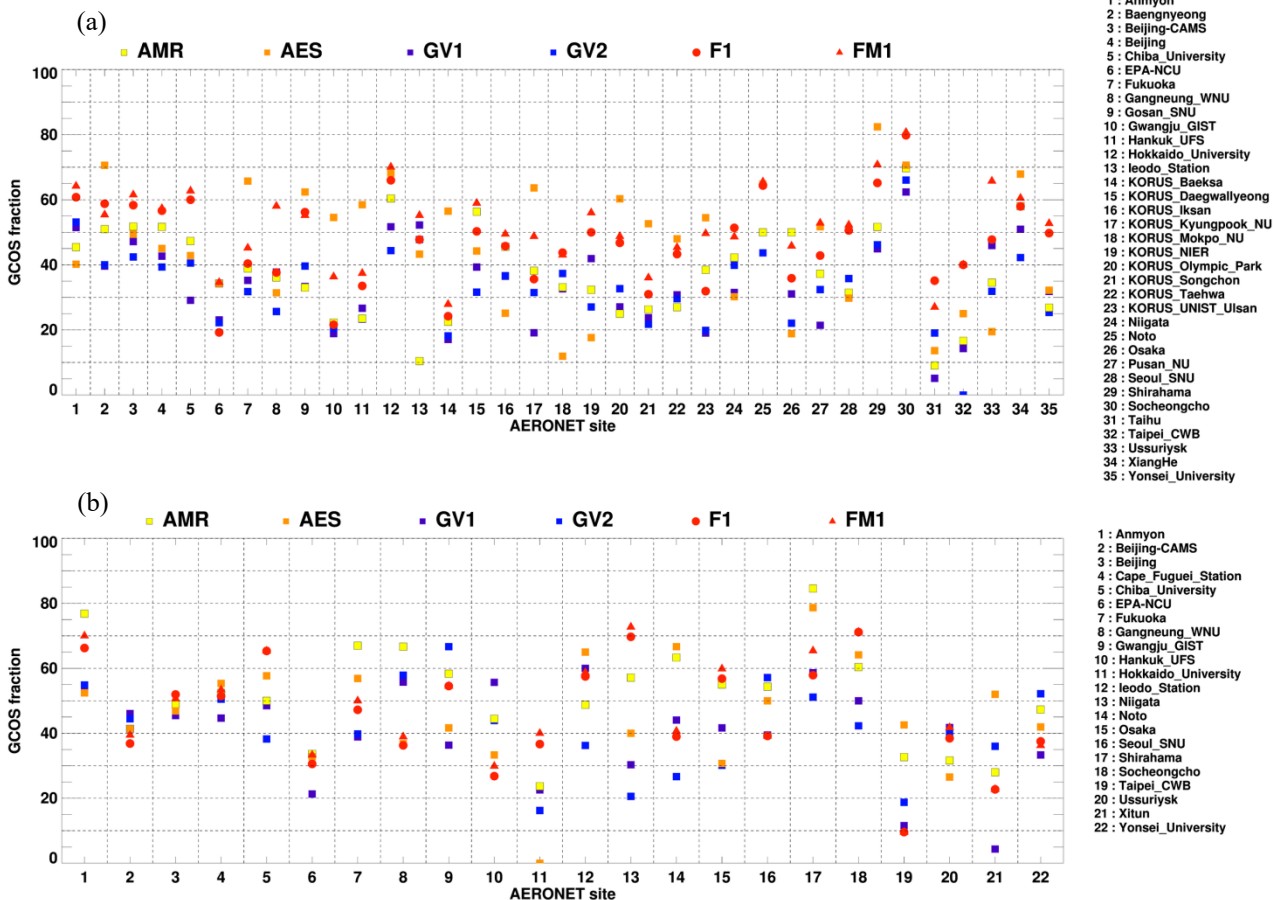

Figure 6. Comparison of the GCOS fraction for respective satellite (AMR, AES, GV1, and GV2), ensemble-mean fusion (F1), and MLE fusion (FM1) during the (a) KORUS-AQ and (b) EMeRGe campaign.

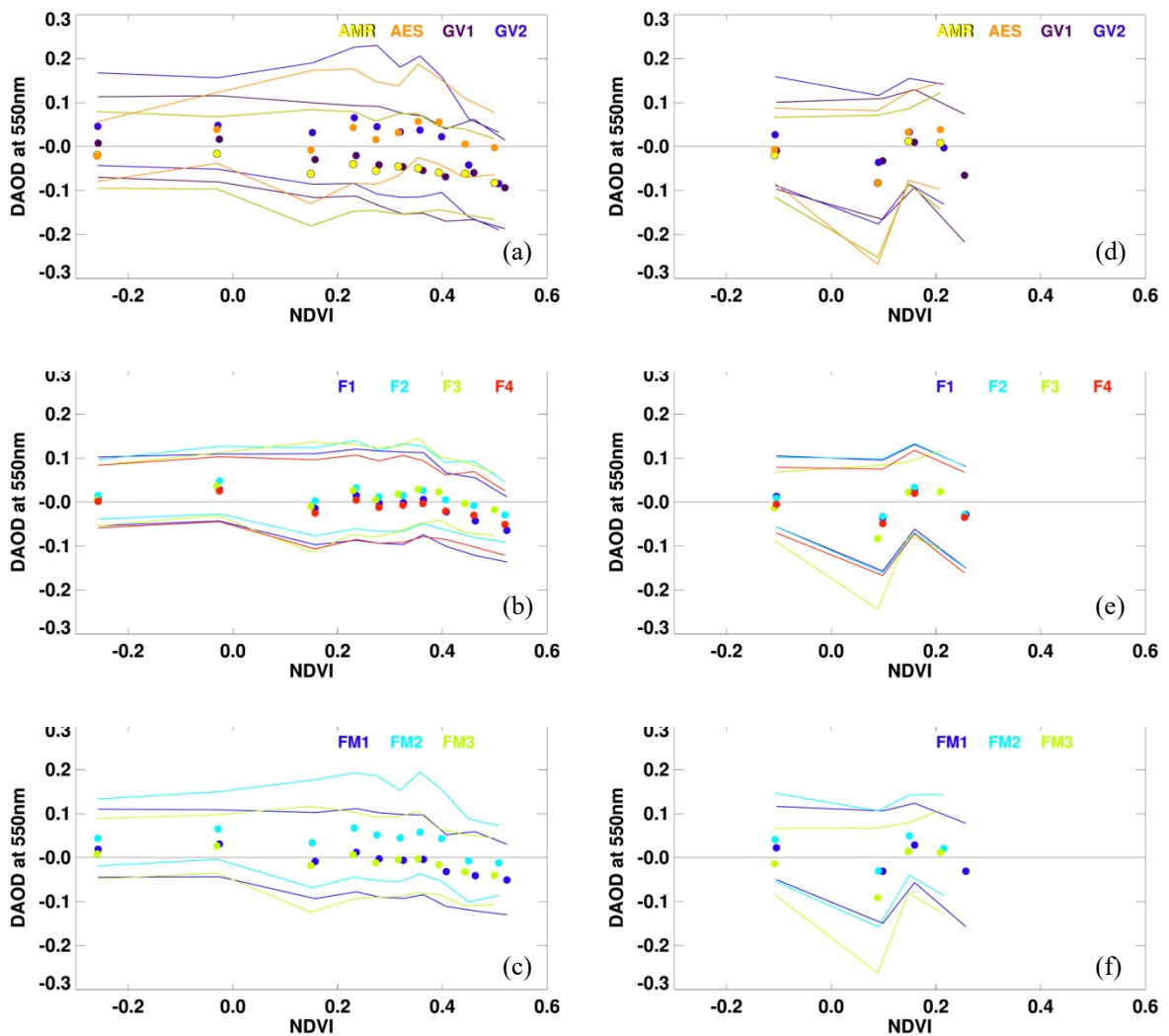

Figure 7. Difference between (a, d) respective, (b, e) ensemble-mean, or (c, f) MLE and AERONET AOD in terms of NDVI during the KORUS-AQ (left column) and the EMeRGe (right column) campaigns. Each points and solid lines represent the median and 1-σ (16$^{th}$ and 84$^{th}$ percentile) of 500 (for the KORUS-AQ) and 400 (for the EMeRGe) collocated data points in terms of NDVI values.

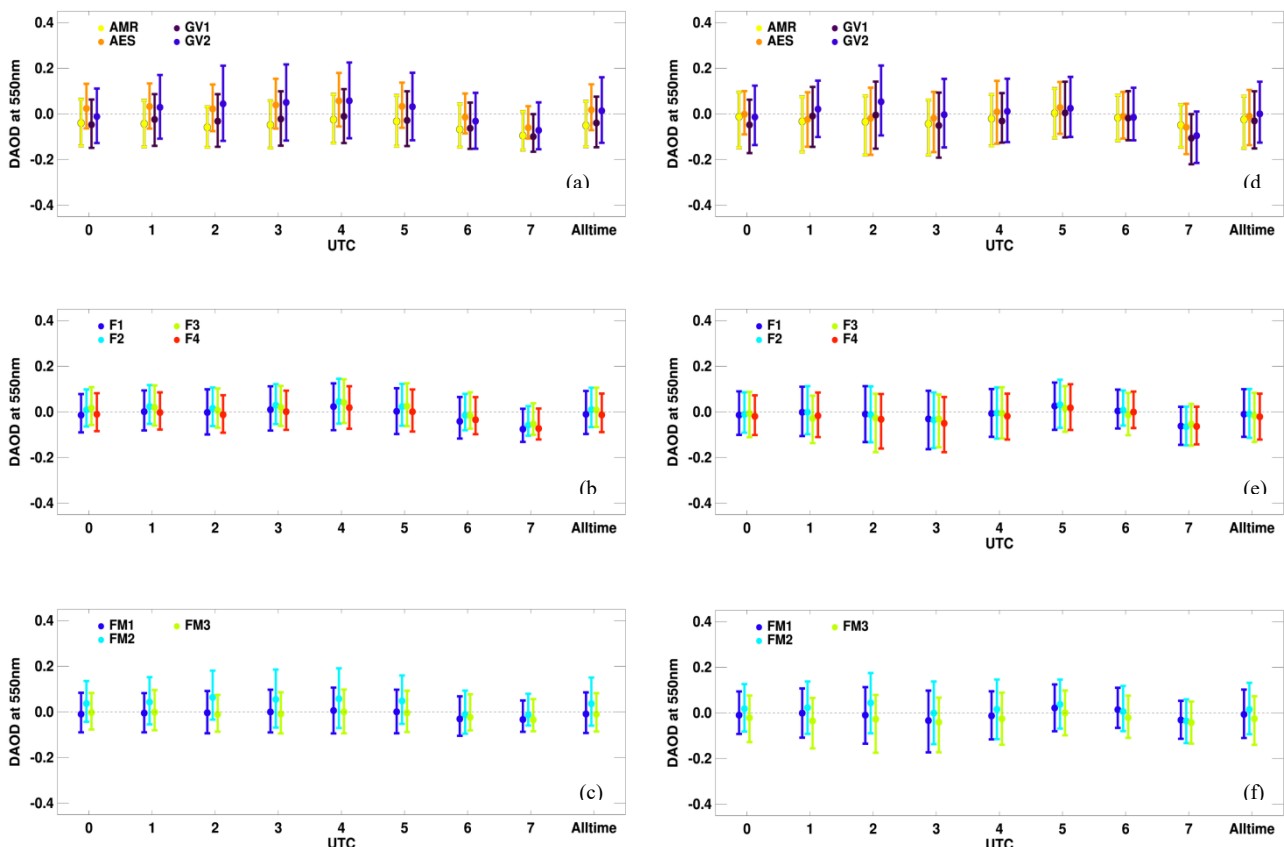

Figure 8. Same as Figure8, but for the observation time.

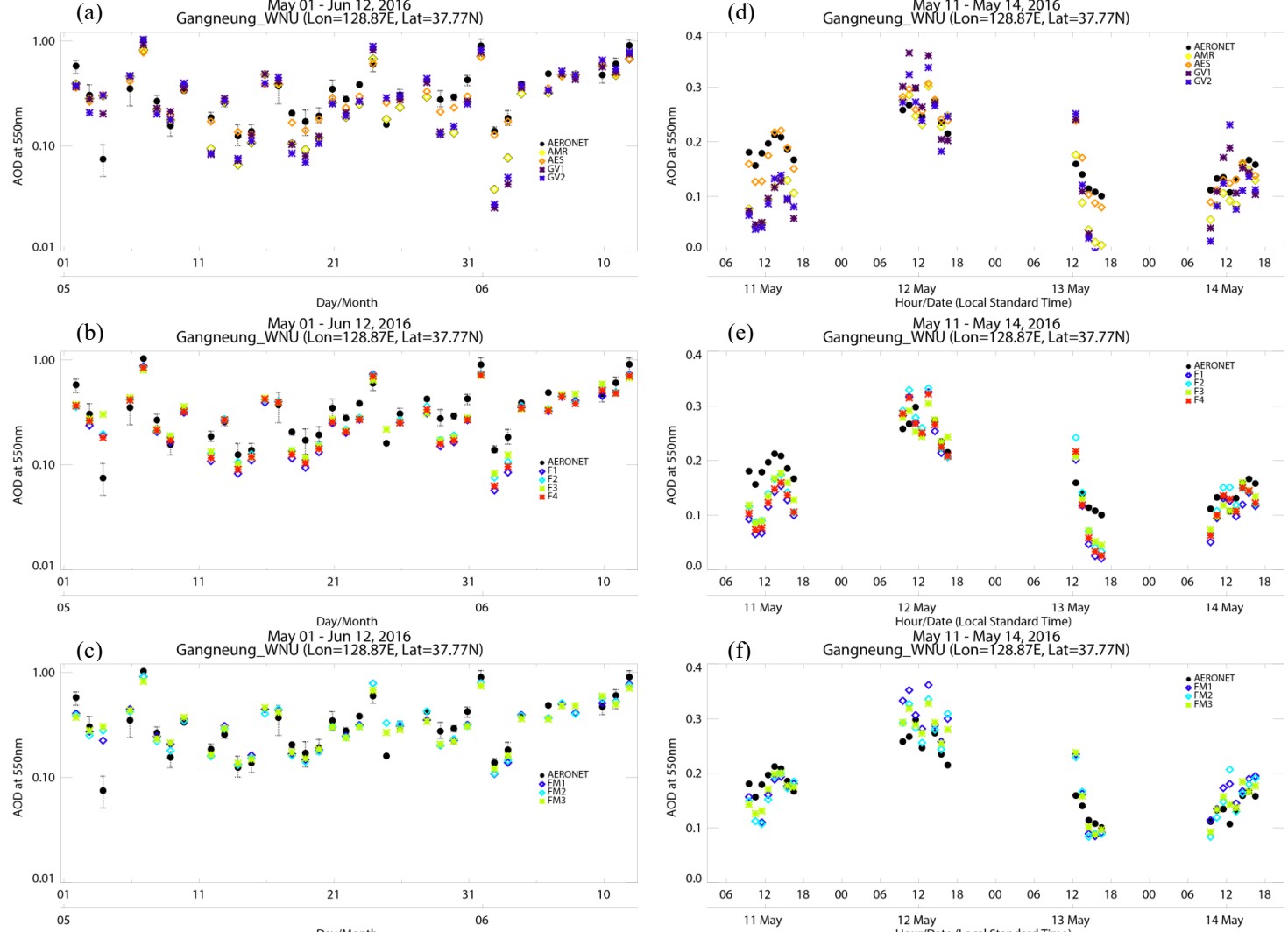

Figure 9. Time series of the daily average AODs at Gangneung WNU site during the KORUS-AQ campaign from (a) respective satellite, (b) ensemble-mean, and (c) MLE fusion. The black-filled circle represents AERONET AOD, and the error bar represents 1-SD of daily AERONET AODs. The diurnal variation in AODs from 11 to 14 May 2016 is shown in the right column, where (d) is the respective satellite, (e) is fused, and (f) is MLE products.

Table 4. Accuracy evaluation of outside of GOCI area of AMR, AES, F3, and FM3 AODs.

| Without GOCI domain | KORUS-AQ AMR | KORUS-AQ AES | KORUS-AQ F3 | KORUS-AQ FM3 | EMeRGe AMR | EMeRGe AES | EMeRGe F3 | EMeRGe FM3 |
|---|---|---|---|---|---|---|---|---|
| N | 1959 | 1958 | 1958 | 1958 | 2610 | 2610 | 2610 | 2610 |
| R | 0.699 | 0.658 | 0.713 | 0.707 | 0.794 | 0.826 | 0.829 | 0.821 |
| RMSE | 0.238 | 0.305 | 0.225 | 0.223 | 0.278 | 0.233 | 0.269 | 0.279 |
| MBE | -0.098 | 0.130 | 0.041 | 0.015 | -0.135 | -0.055 | -0.145 | -0.158 |
| GCOS | 25.6 | 25.6 | 27.3 | 26.5 | 26.8 | 34.1 | 29.0 | 27.5 |

