# Peer review of "Integration of GOCI and AHI Yonsei Aerosol Optical Depth 2 Products During the 2016 KORUS-AQ and 2018 EMeRGe 3 Campaigns"

_Atmospheric Measurement Techniques, 2020_

## Referee Comment (RC1) · Anonymous Referee #1 · 9 Nov 2020

This paper merges and analyzes aerosol optical depth (AOD) data from four data sets (two sensors – AHI and GOCI – each with two different algorithm versions) by two methods (simple mean and maximum likelihood) during two field campaigns in East Asia. Individual and merged data sets are evaluated against Sun photometer observations (more dense than usual due to the field campaigns); statistics of the individual product comparison are also used to inform the merging process for maximum likelihood.

The paper is relevant to the journal and the special issue. The topic is important: we have a lot of satellite AOD data sets now and the question of merging comes up increasingly often. It is also nice to see the geostationary data here; this is a novel aspect

and these new sensors offer temporal coverage unavailable from polar platforms (as the authors point out). So this is all good. The quality of language is ok: the authors have done a good job considering their native languages are not English, but some copy-editing will be required. This can probably be handled by the journal. As a result I have only made language comments when it relates to technical issues.

Some of the analysis is unclear, in particular, relating to the bias correction step (see later comments). I also found the organization of the paper hard to follow: a lot of different merging results were presented but the main message is not clear and I am not sure how well these results could be generalized to other time periods (outside of these field campaigns) or other data sets. Right now it is hard to tell if this is more a paper about these field campaigns, or these retrieval algorithms, or merging in general, because it's not focused/in depth enough. As a result I recommend major revisions to address these issues. My main recommendations relate to streamlining the analysis and discussion, and using more modern merge techniques. I would like to review the revised version. Specific comments in support of my recommendation are below:

1. Line 31: "affect radiative energy" should probably say "affect Earth's radiative energy balance" or "affect solar and thermal radiation" as the current wording feels a little odd.

2. Lines 38-49: there are long citation lists here, with some repetition, and not really much discussion. I suggest consolidating this. We know there are many AOD retrieval algorithms, there's no value in listing a bunch of references unless they are discussed in more detail (as in the examples in the next paragraph). This is an issue elsewhere in the introduction as well, but especially here.

3. Line 50: DT is not one algorithm. It is two algorithms: one for land, one for water. They have the same name, but the assumptions (e.g. aerosol properties, surface reflectance) have nothing in common and even the channels used are different. This should be corrected.

4. Lines 111-125: here the authors describe a number of approaches which have been

used to merge AOD products. Given the sophistication of many of these methods, why are such simple methods (i.e. simple mean, and MLE – which is essentially an uncertainty-weighted mean) used in the present study? Why not use something more state-of-the-art? This paper seems a bit of a missed opportunity to study whether more advanced data fusion approaches as cited in these lines do any better than simple mean or MLE. The authors might consider trying to add a more advanced technique.

5. Section 2: I did not find a clear description of what wavelength(s) AOD is reported at in this analysis. From a few figure captions I think 550 nm, but this seems to be the only mention in the text. This should be stated clearly for each data set used, along with any method for spectral interpolation applied.

6. Line 161: is 0.02 mg/m3 correct? This seems unrealistically low. I was surprised so looked through the Yamada paper cited and did not find this number supported. It looks (e.g. their Figure 2) that most of the time, for their limited domain, the climatological value is 0.1-1 mg/m3. However there is considerable variation. So using 0.02 seems wrong, and having no spatial/monthly variation also seems like it would introduce seasonal biases.

7. Lines 234-244 and Table 1: This is where things get messy for me. I feel there are too many comparisons (7 merge tests, 4 un-merged data sets) and it gets difficult to remember which combinations of algorithm acronym belong to each data merge acronym without going back and forth to the table each time. Further, I am not sure that the split as presented enables the analysis authors want to do. It is complicated because we are splitting between not only different merge types, but also different numbers of sensors (as GOCI has a smaller disk), and also different observation regions (and we know aerosol and surface characteristics, as well as retrieval errors, are probably different in these regions). It is not comparing apples to apples. After reading the paper several times, I'm still not very sure what the message is and how general this recommendation might be. I wonder if it makes more sense to drop some of these experiments and focus only on the ones involving the GOCI disk in order to have a clearer picture for the

analysis (consistent spatial domain, smaller number of comparisons, smaller region to map to make figures easier to read). Maybe doing this, and adding a more advanced merge method (see earlier comment), would give an analysis which is easier to follow and of broader interest. Having all the 11-panel figures which look mostly quite similar is hard to follow.

8. Section 3.4: this section doesn't seem to actually explain how the bias correction was done. More detail is needed. Also, I don't see the evidence that retrieval errors do follow a Gaussian distribution: there is no Gaussian distribution comparison shown in Figure 1. This could be demonstrated better by e.g. a QQ plot. Further, it could be that there are multiple populations in here and it looks reasonably Gaussian on average, but not for subsets of the data.

9. Line 264: Sayer (2013) does not show AOD follows a lognormal distribution. Perhaps the authors are thinking of Sayer and Knobelspiesse (2019)? https://acp.copernicus.org/articles/19/15023/2019/

10. Sections 4, 5: these mostly just describe the figures and again, because there's a lot of maps and scatter plots which look very similar, it is hard to pick out the main message. This supports my idea to pick which experiments and parts of the data are most important and focus on those. In my view the figures should support the text; the text should clearly offer explanations and recommendations and not just describe the figures. I don't have many more specific comments on these for that reason.

11. Tables 2, 3: these are a bit of a sea of numbers. It is hard for the reader to parse them and extract the main message. If the variation between entries is important, perhaps these should be figures instead. Also, "NaN" does not belong in a table like this. If there were no data, leave it blank or put a "-". NaN is computer code.

12. Figures 5, 7: I recommend the regression fits be removed here. As the authors note, AOD is close to lognormal. Also, the AOD error is dependent on AOD. Also, the fact that there are NDVI dependences of retrieval errors means that there are multiple

populations of data with distinct characteristics here. All this means that the regression used is statistically inappropriate. It should be removed in order for the paper to be correct. I do not believe the regressions are vital to the discussion anyway.

13. Figure 9: can uncertainty bars be added here? It is hard to see whether these differences are real or within sampling error. Also, the x axis should be checked. While NDVI below zero is mathematically possible, it is not realistic except for water bodies or cloud-contaminated pixels. I am surprised that values seem to vary between -0.3 and +0.4 or so. Even deserts have an NDVI around 0.1-0.2, and vegetated areas often above 0.5. I wonder if there is perhaps a bug, a definition difference, or a serious spectral error in the surface reflectance model producing these values. This should be checked.

14. Figure 10: this has vertical bars but they are not explained. Is this standard deviation, standard error, or something else?

15. Conceptually, I also have an issue with using AERONET to train a bias correction and then evaluating the bias-corrected data against AERONET. Of course this will look better than the original products. I am not sure of the best way around this though. Again, streamlining the number of comparisons made in the paper will make it more readable and allow a better understanding of the advantages of the methods.

---

## Referee Comment (RC2) · Anonymous Referee #2 · 9 Nov 2020

GENERAL COMMENTS ——————

This article describes a study to compare different techniques for fusing aerosol optical depth products from two multi-spectral instruments viewing East Asia from geostationary orbit, GOCI and AHI, and evaluate the results during two field campaign periods. The topic is relevant given that these two instruments represent current state-of-the-art capabilities for diurnal aerosol observations from satellites. The hypothesis is that some type of ensemble mean will usually perform better than any individual member, and the paper compares 4 individual retrievals (2 each from GOCI and AHI), 4 different simple ensemble-mean combinations, and 3 different maximum-likelihood-

estimate (MLE) combinations. In general, the hypothesis seems true, with the fused products generally overcoming different deficiencies in the individual products. However, the number of different permutations considered makes it difficult to focus on what attributes lead to the best improvements. The impact of bias correction should be clarified. Also, the current multi-panel figures make it difficult to see the differences. This work would make a valuable contribution to the literature if the clarity of the presentation can be improved. Specific suggestions are offered as follows.

SPECIFIC COMMENTS ──────────────

The discussion of gap filling techniques starting at line 115 needs an introduction to provide context for the geostationary observations. While the need for gap filling in daily LEO observations is somewhat intuitive, it seems the simplest fusion of GEO products could produce a high yield without gap filling. Please clarify how the gap-filling applies to the current work.

The organization of sections 2 and 3 was quite confusing to me; I had to re-read several times to understand what was being done. 2.1 is fine, simply describing the two AHI products. The first paragraph of 2.2 is fine, simply describing the two GOCI products. The second paragraph (beginning line 195) belongs in a separate section describing the different fusions, rather than in the GOCI algorithm section. Section 2.3 seems out of place; I suggest it be moved such that it is the last text before the Results section. Section 3 would greatly benefit from starting with a simple statement of the approach, e.g., we compare 4 different simple ensemble-mean combinations and 3 different MLE combinations. It is confusing to read about the MLE fusions in the Ensemble Mean section 3.2 (lines 242-244); I suggest you describe the FM entries of Table 1 in the next section about MLE method, instead of in this section about ensemble mean method. And at that point, note that the same members are included in F2 and FM2, F3 and FM3, F4 and FM1, as can be seen in Table 1.

Further, I suggest you swap F1 and F4, so that the same members are included in

F1 and FM1; this would make it easier for the reader to examine the differences in the figures. For example, adjacent panels (e) and (i) would be for the same members, similar to how adjacent panels (f,j) and (g,k) are for the same members, in Figs 2-3 and 5-8.

Lines 256-261 (calculation of RMSE values) are confusing to me. If I understand correctly what you are doing, this could be explained much more clearly as follows. The locations of ground measurements are very sparse in comparison with the satellite coverage, so you choose to model RMSE as a function of NDVI. Then you bin all ground/satellite co-locations with respect to NDVI, AOD, and time, calculate the RMSE in each bin, and then apply this RMSE to every satellite pixel as a table lookup based on those 3 parameters. A similar comment applies to the description of bias correction technique (section 3.4, lines 267-272).

It appears that bias correction is applied to the MLE fusions but not to the simple-mean fusions. It seems that bias-correcting the simple-mean fusions could easily be done and would provide a more direct comparison of the two techniques. And if you do this, you should be able to say something about the importance of bias correction in isolation.

The panels in figures 2, 3, 6, and 8 are so small that it is very difficult to see the differences in any features. Also, the large-domain view is only relevant for 4 cases (AER, AMR, F3, FM3). I suggest splitting each of these figures into 2 figures. The first would be for the large domain and would only have 4 panels, so the panels could all be twice as large. The second would be for the small (GOCI) domain and although it would have the same number of panels, there would be much less wasted white space since you would be zoomed into the GOCI region. Even with these changes, it may be difficult for the reader to actually see the differences among the cases. Consider selecting one product to show a representative AOD distribution (perhaps the one you consider to be the "best") and plot all other cases as the difference from this reference; this would allow you to clearly highlight where the differences arise.
It is very hard to see the differences in Figure 4. I suggest the figure could be greatly improved by plotting AERONET and DAOD (the difference from AERONET) for all the other products. The left vertical axis could be AOD and the right vertical axis could be DAOD. I don't think you would need a log scale.

Line 335: The statement "the fusion products have a value of 0.131, lower than the minimum value of various satellite products (0.161)" is not true for all cases; Fused3 is .163, essentially the same as the .161 value quoted as the minimum of the individual products.

Paragraph beginning on line 349: This discussion highlights what I see as a problem with the current analysis. The AHI-only fusion results could also be analyzed within the small (GOCI) domain. Then there would be direct comparisons of all the ensembles in the small domain and separate comparisons of the large-domain results. I noted that Figure 11 does do this, which is the best aspect of Fig 11, and find it strange that this isn't done more consistently throughout the analysis.

First paragraph of section 5.2: AHI_ESR is in all the ensembles, so it is not surprising that the ensemble results move toward the AHI_ESR behavior. It explains the KORUS-AQ results; AHI_ESR has a positive bias, the other 3 have negative or no bias, so the combinations will produce small biases. Similar behavior is seen in EMeRGe; the results tend to collapse toward the original AHI_ESR values.

Discussion of Fig 11, in particular paragraph beginning on line 394: It is really impossible to see that "results after fusion show slightly better than respective satellite product accuracy in terms of SD, RMSE, and EE values" since so many points are clustered essentially on top of each other. All that can be clearly seen is that GV1, GV2, and the "all" points are distinct from the cluster of everything else. It appears that the best discriminator between points in the cluster is the %EE. Two suggestions could improve this figure. First, in the legend, color the symbols by the %EE for each case, then at least it will be easy to see that it improves for the ensembles. Second, consider adding

an inset that zooms in on the cluster of points. This may be irrelevant though (and Fig 11 redundant), if the result is that there are only minimal differences between the various cases.

Line 410: This could also be because EMeRGe was in a period of brighter surface reflectance.

Summary and Conclusion: I think it is important to point out that the ensemble-mean and MLE techniques produce very similar results, based on the numbers in Figures 5 and 7.

Sentence on lines 433-435: This appears to be marginally true for EMeRGe (fig 7) but is not true for KORUS-AQ (fig 5), where ensemble-mean actually appears to be best.

TECHNICAL CORRECTIONS ──────────────────

The citations on lines 48-49 duplicate citations earlier in the paragraph.

Line 195: unclear meaning; instead of the word "for", do you mean "depending on"?

Line 197: Not sure what is meant by "the NDVI shows a negative bias". Isn't NDVI an independent variable? Do you mean, when AOD is analyzed as a function of NDVI, a low bias exists for all values of NDVI?

Throughout, be consistent between the convention used in the text, tables, and figures. The text mostly uses the "short" labels e.g. as defined in lines 231-232, while the figures use a "long" convention that is much easier for the reader to keep straight (e.g., AHI_MRM instead of AMR). As a reader, I preferred the longer conventions because of this ease of keeping things straight.

Line 240: define what is meant by "wide area". It becomes clear when looking at the figures, however at this point in the paper it would be helpful to define it.

Line 266: Typo, AESR should be AES

Line 382: Start a new paragraph for the discussion of Figure 11.

Throughout the manuscript, be consistent with terminology. Fusion is the generic term. Two fusion techniques are used, ensemble mean and MLE. E.g., the legend in Fig 11 should indicate ensemble-mean rather than fusion.

Line 429-430: It seems the biases are not "due to" NDVI etc., but instead are represented as functions of NDVI, time of day, and AOD.

Tables 1 and 3: The NDVI labels are identical for the 2nd and 3rd groupings. It seems one or both are typographical errors.

———————————————————

---

## Author Comment (AC1) · 15 Dec 2020

This paper merges and analyzes aerosol optical depth (AOD) data from four data sets (two sensors – AHI and GOCI – each with two different algorithm versions) by two methods (simple mean and maximum likelihood) during two field campaigns in East Asia. Individual and merged data sets are evaluated against Sun photometer observations (more dense than usual due to the field campaigns); statistics of the individual product comparison are also used to inform the merging process for maximum likelihood. The paper is relevant to the journal and the special issue. The topic is important: we have a lot of satellite AOD data sets now and the question of merging comes up increasingly often. It is also nice to see the geostationary data here; this is a novel aspect and these new sensors offer temporal coverage unavailable from polar platforms (as the authors point out). So this is all good. The quality of language is ok: the authors have done a good job considering their native languages are not English, but some copy-editing will be required. This can probably be handled by the journal. As a result I have only made language comments when it relates to technical issues.

Some of the analysis is unclear, in particular, relating to the bias correction step (see later comments). I also found the organization of the paper hard to follow: a lot of different merging results were presented but the main message is not clear and I am not sure how well these results could be generalized to other time periods (outside of these field campaigns) or other data sets. Right now it is hard to tell if this is more a paper about these field campaigns, or these retrieval algorithms, or merging in general, because it's not focused/in depth enough. As a result I recommend major revisions to address these issues. My main recommendations relate to streamlining the analysis and discussion, and using more modern merge techniques. I would like to review the revised version. Specific comments in support of my recommendation are below:

1. Line 31: "affect radiative energy" should probably say "affect Earth's radiative energy balance" or "affect solar and thermal radiation" as the current wording feels a little odd.

*- Thank you for your comment. We revised this sentence.*

2. Lines 38-49: there are long citation lists here, with some repetition, and not really much discussion. I suggest consolidating this. We know there are many AOD retrieval algorithms, there's no value in listing a bunch of references unless they are discussed in more detail (as in the examples in the next paragraph). This is an issue elsewhere in the introduction as well, but especially here.

*- The inserted reference in the sentence was deleted.*

3. Line 50: DT is not one algorithm. It is two algorithms: one for land, one for water. They have the same name, but the assumptions (e.g. aerosol properties, surface reflectance) have nothing in common and even the channels used are different. This should be corrected.

*- Thank you for your comment. However, the DT-ocean and DT-land algorithms are often referred to as DT algorithms. Each algorithm's characteristics (land and ocean) are briefly mentioned in line 53-59.*

4. Lines 111-125: here the authors describe a number of approaches which have been used to merge AOD products. Given the sophistication of many of these methods, why are such simple methods (i.e. simple mean, and MLE – which is essentially an uncertainty-weighted mean) used in the present study? Why not use something more state-of-the-art? This paper seems a bit of a missed opportunity to study whether more advanced data fusion approaches as cited in these lines do any better than simple mean or MLE. The authors might consider trying to add a more advanced technique.

*- Previous studies mentioned in this paper include data fusion based on Kriging, reproduction of spectral AOD, and BME method. Most of them focus on gap filling and rebuild AOD in areas not observed by MISR, MODIS, and SeaWiFS, and so on (Wang et al., 2013; Tang et al., 2016). Here we focus on a study that attempts to improve the accuracy of AOD products at the retrieved pixels, thus shows the ensemble mean and MLE fusion, respectively, to compare these two, one very simple one and the other with more elaborated processes. Because the previous studies on AOD*

*fusion improved the retrieved values mainly based on MLE or NDVI-based fusion studies (Wei et al., 2019, Levy et al., 2013), we tried to further improve them with rather simple approach to save computation time considering the nature of satellite product file size and user's near-real-time demand for data assimilation. Compared to the AERONET, the MLE method improved the scattered results of the satellite AOD, but did not correct the systematic bias, so additional bias correction was performed.*

*In addition, most of the fusion methods do not consider the uncertainties in each AOD product used, especially the uncertainty in the pixel scale. While some fusion algorithms do consider the uncertainty of source data, they rarely consider the systematic error of the product itself when calculating the uncertainty (Xu et al., 2015; Xie et al., 2018).*

5. Section 2: I did not find a clear description of what wavelength(s) AOD is reported at in this analysis. From a few figure captions I think 550 nm, but this seems to be the only mention in the text. This should be stated clearly for each data set used, along with any method for spectral interpolation applied.

*- Thank you for your comment. We added the wavelength information on Line 134, 150, 208 (revised manuscript).*

6. Line 161: is 0.02 mg/m3 correct? This seems unrealistically low. I was surprised so looked through the Yamada paper cited and did not find this number supported. It looks (e.g. their Figure 2) that most of the time, for their limited domain, the climatological value is 0.1-1 mg/m3. However there is considerable variation. So using 0.02 seems wrong, and having no spatial/monthly variation also seems like it would introduce seasonal biases.

*- Sorry for confusing.   We removed reference.*

*0.02 represents the average climate value over clean ocean, and the sentence was revised.*

*In addition, 0.02mg/m$^3$ used in the AHI ESR method was used only for CHL pixels that were not retrieved by the JAXA algorithm(Murakami. 2016), and according to Lim et al. (Remote Sensing, 2018), the maximum AOD error according to the CHL-a concentration of 50 mg/m3 in the YAER algorithm was 0.08.*

7. Lines 234-244 and Table 1: This is where things get messy for me. I feel there are too many comparisons (7 merge tests, 4 un-merged data sets) and it gets difficult to remember which combinations of algorithm acronym belong to each data merge acronym without going back and forth to the table each time. Further, I am not sure that the split as presented enables the analysis authors want to do. It is complicated because we are splitting between not only different merge types, but also different numbers of sensors (as GOCI has a smaller disk), and also different observation regions (and we know aerosol and surface characteristics, as well as retrieval errors, are probably different in these regions). It is not comparing apples to apples. After reading the paper several times, I'm still not very sure what the message is and how general this recommendation might be. I wonder if it makes more sense to drop some of these experiments and focus only on the ones involving the GOCI disk in order to have a clearer picture for the analysis (consistent spatial domain, smaller number of comparisons, smaller region to map to make figures easier to read). Maybe doing this, and adding a more advanced merge method (see earlier comment), would give an analysis which is easier to follow and of broader interest. Having all the 11-panel figures which look mostly quite similar is hard to follow.

*- Thank you for your comment. FM1 (MLE all) was selected as the representative fused AOD, and the domain area was reduced to GOCI's coverage. Other products were shown as differences from the FM1. Thanks to this update, the difference in fusion is well expressed.*

8. Section 3.4: this section doesn't seem to actually explain how the bias correction was done. More detail is needed. Also, I don't see the evidence that retrieval errors do follow a Gaussian distribution: there is no Gaussian distribution comparison shown in Figure 1. This could be demonstrated better by e.g. a QQ plot. Further, it could be that there are multiple populations in here and it looks reasonably Gaussian on average, but not for subsets of the data.

*- Thank you for your comment. We added more detail in the bias correction in section 3.4 and revised the Q-Q plot.*

9. Line 264: Sayer (2013) does not show AOD follows a lognormal distribution. Perhaps the authors are thinking of Sayer and Knobelspiesse (2019)? https://acp.copernicus.org/articles/19/15023/2019/

*- Sorry for the confusion. We revised with the reference given by the reviewer's comment.*

10. Sections 4, 5: these mostly just describe the figures and again, because there's a lot of maps and scatter plots which look very similar, it is hard to pick out the main message. This supports my idea to pick which experiments and parts of the data are most important and focus on those. In my view the figures should support the text; the text should clearly offer explanations and recommendations and not just describe the figures. I don't have many more specific comments on these for that reason.

*- Thank you for your comment. We added and revised the figures, texts, and revised the conclusion.*

11. Tables 2, 3: these are a bit of a sea of numbers. It is hard for the reader to parse them and extract the main message. If the variation between entries is important, perhaps these should be figures instead. Also, "NaN" does not belong in a table like this. If there were no data, leave it blank or put a "-". NaN is computer code.

*- Thank you for your comment. According to your comment, both Tables 2 and 3 were replaced with Figures 1 and 3. The period (2018.04-2019.03, but excluding the EMeRGe campaign) was modified for statistical analysis to avoid data redundancy.*

12. Figures 5, 7: I recommend the regression fits be removed here. As the authors note, AOD is close to lognormal. Also, the AOD error is dependent on AOD. Also, the fact that there are NDVI dependences of retrieval errors means that there are multiple populations of data with distinct characteristics here. All this means that the regression used is statistically inappropriate. It should be removed in order for the paper to be correct. I do not believe the regressions are vital to the discussion anyway.

*- Thank you for your comment. We revised Figures 7 and 9.*

13. Figure 9: can uncertainty bars be added here? It is hard to see whether these differences are real or within sampling error. Also, the x axis should be checked. While NDVI below zero is mathematically possible, it is not realistic except for water bodies or cloud-contaminated pixels. I am surprised that values seem to vary between -0.3 and +0.4 or so. Even deserts have an NDVI around 0.1-0.2, and vegetated areas often above 0.5. I wonder if there is perhaps a bug, a definition difference, or a serious spectral error in the surface reflectance model producing these values. This should be checked.

*- For the collocation with AERONET, satellite AODs within 25km is averaged, which tends to decrease values partly, but it is confirmed that the maximum NDVI value is about 0.7. Also, negative NDVI appears because the ocean pixel (AERONET near coastal) is included. It may also look somewhat low because the average was taken as the representative value of the collocation points and plotted. The below figure shows the collocated NDVI values during each campaign period.*

[Figure]

14. Figure 10: this has vertical bars but they are not explained. Is this standard deviation, standard error, or something else?

*- Sorry for missing this. We added information of vertical bar, which was to show 1-sigma.*

15. Conceptually, I also have an issue with using AERONET to train a bias correction and then evaluating the bias-corrected data against AERONET. Of course this will look better than the original products. I am not sure of the best way around this though. Again, streamlining the number of comparisons made in the paper will make it more readable and allow a better understanding of the advantages of the methods.

*- The bias correction and RMSE were calculated using data for about one year from April 2018 to March 2019 (excluding the EMeRGe period) to avoid redundancy of all data. Therefore, we revised the results deviating from the cyclical logic that the reviewer told us because the error analysis was performed using the algorithm's characteristics for one year that does not overlap with a specific period.*

---

## Author Comment (AC2) · 15 Dec 2020

**Responses to Reviewer's Comments:**

*We appreciate the reviewer's comments and suggestions, which were very helpful in improving the overall quality of our manuscript. Basically, all the comments and suggestions were reflected in our revision. Our responses are listed below to each comment.*

This article describes a study to compare different techniques for fusing aerosol optical depth products from two multi-spectral instruments viewing East Asia from geostationary orbit, GOCI and AHI, and evaluate the results during two field campaign periods. The topic is relevant given that these two instruments represent current state-of-the-art capabilities for diurnal aerosol observations from satellites. The hypothesis is that some type of ensemble mean will usually perform better than any individual member, and the paper compares 4 individual retrievals (2 each from GOCI and AHI), 4 different simple ensemble-mean combinations, and 3 different maximum-likelihood-estimate (MLE) combinations. In general, the hypothesis seems true, with the fused products generally overcoming different deficiencies in the individual products. However, the number of different permutations considered makes it difficult to focus on what attributes lead to the best improvements. The impact of bias correction should be clarified. Also, the current multi-panel figures make it difficult to see the differences. This work would make a valuable contribution to the literature if the clarity of the presentation can be improved. Specific suggestions are offered as follows.

SPECIFIC COMMENTS ────────────────

The discussion of gap filling techniques starting at line 115 needs an introduction to provide context for the geostationary observations. While the need for gap filling in daily LEO observations is somewhat intuitive, it seems the simplest fusion of GEO products could produce a high yield without gap filling. Please clarify how the gap-filling applies to the current work.

*- As reviewers commented, retrievals and applications using geostationary satellite observations cover many areas. Sorry for the lack of clarity in our originally submitted manuscript. Our aim was to provide optimized aerosol products from two different algorithm and two different instruments (GOCI and AHI). Therefore, this paper aims to produce the optimal fused AOD products where retrieved results are available, not to fill the gap where aerosol properties are not retrieved.*

The organization of sections 2 and 3 was quite confusing to me; I had to re-read several times to understand what was being done. 2.1 is fine, simply describing the two AHI products. The first paragraph of 2.2 is fine, simply describing the two GOCI products. The second paragraph (beginning line 195) belongs in a separate section describing the different fusions, rather than in the GOCI algorithm section.

*- Thank you for your comment. This part has been moved to the beginning of section 3.*

Section 2.3 seems out of place; I suggest it be moved such that it is the last text before the Results section. Section 3 would greatly benefit from starting with a simple statement of the approach, e.g., we compare 4 different simple ensemble-mean combinations and 3 different MLE combinations. It is confusing to read about the MLE fusions in the Ensemble Mean section 3.2 (lines 242-244); I suggest you describe the FM entries of Table 1 in the next section about MLE method, instead of in this section about ensemble mean method. And at that point, note that the same members are included in F2 and FM2, F3 and FM3, F4 and FM1, as can be seen in Table 1. Further, I suggest you swap F1 and F4, so that the same members are included in F1 and FM1; this would make it easier for the reader to examine the differences in the figures. For example, adjacent panels (e) and (i) would be for the same members, similar to how adjacent panels (f,j) and (g,k) are for the same members, in Figs 2-3 and 5-8.

*- Thanks for your suggestion to reorganize, which improves the manuscript story flow. In the next session, we revised our manuscript to mention FM1-3, and moved Sec 2.3 to the last paragraph before the results section. Also, we swapped F1 and F4, per reviewer's suggestion to improve readability.*

Lines 256-261 (calculation of RMSE values) are confusing to me. If I understand correctly what you are doing, this could be explained much more clearly as follows. The locations of ground measurements are very sparse in comparison with the satellite coverage, so you choose to model RMSE as a function of NDVI. Then you bin all ground/satellite co-locations with respect to NDVI, AOD, and time, calculate the RMSE in each bin, and then apply this RMSE to every satellite pixel as a table lookup based on those 3 parameters.

*- Thank you for your advice. We revised these sentences.*

A similar comment applies to the description of bias correction technique (section 3.4, lines 267-272). It appears that bias correction is applied to the MLE fusions but not to the simple-mean fusions. It seems that bias-correcting the simple-mean fusions could easily be done and would provide a more direct comparison of the two techniques. And if you do this, you should be able to say something about the importance of bias correction in isolation.

*- The simple average fusion field is an ensemble averaging technique, which utilizes the characteristic of finding a better value when multiple signals are averaged. In the main texts, we would like to mention that the accuracy becomes better (the less scattered), as we have more ensemble members. Our purpose was to show how well they matched the MLE fusion products through bias correction and pixel-based error fusion. However, to demonstrate the comments pointed out by the reviewers, the result of performing the bias correction is attached below*

[Figure]

**Figure 1. KORUS-AQ campaign**

[Figure]

**Figure 2. EMeRGe campaign.**

*- Looking at the effect of bias correction, F1 using all outputs shows an improvement results. Meanwhile, F4, which uses an ensemble member similar to F1, decreases in KORUS-AQ and increases %EE in the EMeRGe campaign. This may appear because the GV1's bias correction value is not accurate. Although mentioned in the text, the accurate correction may not be made using the RMSE and bias correction in this study for long-term analysis values. In general, if bias correction is performed and ensemble averaging is performed, MBE is improved in most cases, but the difference in EMeRGe F3 product is the greatest.*

The panels in figures 2, 3, 6, and 8 are so small that it is very difficult to see the differences in any features. Also, the large-domain view is only relevant for 4 cases (AER, AMR, F3, FM3). I suggest splitting each of these figures into 2 figures. The first would be for the large domain and would only have 4 panels, so the panels could all be twice as large. The second would be for the small (GOCI) domain and although it would have the same number of panels, there would be much less wasted white space since you would be zoomed into the GOCI region. Even with these changes, it may be difficult for the reader to actually see the differences among the cases. Consider selecting one product to show a representative AOD distribution (perhaps the one you consider to be the "best") and plot all other cases as the difference from this reference; this would allow you to clearly highlight where the differences arise.

*- Thank you for your suggestion. We revised the Figure 4, 5 and paragraphs with one representative average AOD (FM1), while the remaining products were modified to show*

*differences, mean (XX) – mean (FM1), for the same area as the GOCI's.*

It is very hard to see the differences in Figure 4. I suggest the figure could be greatly improved by plotting AERONET and DAOD (the difference from AERONET) for all the other products. The left vertical axis could be AOD and the right vertical axis could be DAOD. I don't think you would need a log scale.

*- Thank you for your suggestion. DAOD was added as the reviewer suggested. However, if the figure is not shown in log-scale, the variation of the low AOD does not appear well, so the symbol thickness has been modified to be thin.*

Line 335: The statement "the fusion products have a value of 0.131, lower than the minimum value of various satellite products (0.161)" is not true for all cases; Fused3 is .163, essentially the same as the .161 value quoted as the minimum of the individual products.

*- Sorry for the confusion. We corrected sentence based on the re-calculated results, and added the phrase excluding F3, and FM3.*

Paragraph beginning on line 349: This discussion highlights what I see as a problem with the current analysis. The AHI-only fusion results could also be analyzed within the small (GOCI) domain. Then there would be direct comparisons of all the ensembles in the small domain and separate comparisons of the large-domain results. I noted that Figure 11 does do this, which is the best aspect of Fig 11, and find it strange that this isn't done more consistently throughout the analysis.

*- Thanks for the suggestion. As mentioned above, analysis domain was set as the GOCI domain in Figure 4 and 5. And, we added to section 5.3 and table 2. This section and table 2 were shown two AHI AODs validation score within GOCI's observation area.*

First paragraph of section 5.2: AHI_ESR is in all the ensembles, so it is not surprising that the ensemble results move toward the AHI_ESR behavior. It explains the KORUS-AQ results; AHI_ESR has a positive bias, the other 3 have negative or no bias, so the combinations will produce small biases. Similar behavior is seen in EMeRGe; the results tend to collapse toward the original

AHI_ESR values. Discussion of Fig 11, in particular paragraph beginning on line 394: It is really impossible to see that "results after fusion show slightly better than respective satellite product accuracy in terms of SD, RMSE, and EE values" since so many points are clustered essentially on top of each other.

All that can be clearly seen is that GV1, GV2, and the "all" points are distinct from the cluster of everything else. It appears that the best discriminator between points in the cluster is the %EE. Two suggestions could improve this figure. First, in the legend, color the symbols by the %EE for each case, then at least it will be easy to see that it improves for the ensembles. Second, consider adding an inset that zooms in on the cluster of points. This may be irrelevant though (and Fig 11 redundant), if the result is that there are only minimal differences between the various cases.

*- Thank you for your suggestion. The legend is also shown as %EE, and only all site validation is shown in the Taylor diagram as suggested by the reviewer. The validation of broader products collocated with GOCI is summarized in Table 2.*

Line 410: This could also be because EMeRGe was in a period of brighter surface reflectance.

*- Thank you for your comments. We added this point per the reviewer's comment.*

Summary and Conclusion: I think it is important to point out that the ensemble-mean and MLE techniques produce very similar results, based on the numbers in Figures 5 and 7.

*- Thank you for your comment. We added this point per the reviewer's comment.*

Sentence on lines 433-435: This appears to be marginally true for EMeRGe (fig 7) but is not true for KORUS-AQ (fig 5), where ensemble-mean actually appears to be best.

*- Thank you for your comment. We revised this sentence.*

TECHNICAL CORRECTIONS ————————————

The citations on lines 48-49 duplicate citations earlier in the paragraph.

*- We removed duplicate citations.*

Line 195: unclear meaning; instead of the word "for", do you mean "depending on"?

*- We revised this word.*

Line 197: Not sure what is meant by "the NDVI shows a negative bias". Isn't NDVI an independent variable? Do you mean, when AOD is analyzed as a function of NDVI, a low bias exists for all values of NDVI?

*- Sorry for the confusion. We revised this sentence.*

Throughout, be consistent between the convention used in the text, tables, and figures. The text mostly uses the "short" labels e.g. as defined in lines 231-232, while the figures use a "long" convention that is much easier for the reader to keep straight (e.g., AHI_MRM instead of AMR). As a reader, I preferred the longer conventions because of this ease of keeping things straight.

*- Thank you for your comments. We revised that the abbreviations of the text and pictures have been unified.*

Line 240: define what is meant by "wide area". It becomes clear when looking at the figures, however at this point in the paper it would be helpful to define it.

*- Thank you for your suggestion. We added domain information.*

Line 266: Typo, AESR should be AES

*- Thank you for your comment. We removed this sentence.*

Line 382: Start a new paragraph for the discussion of Figure 11.

*- Thank you for your suggestion. We revised.*

Throughout the manuscript, be consistent with terminology. Fusion is the generic term. Two fusion techniques are used, ensemble mean and MLE. E.g., the legend in Fig 11 should indicate ensemble-mean rather than fusion.

*- Thank you for your comment. We revised the all of the legends.*

Line 429-430: It seems the biases are not "due to" NDVI etc., but instead are represented as functions of NDVI, time of day, and AOD.

*- Thank you for your comments. We revised this sentence.*

Tables 1 and 3: The NDVI labels are identical for the 2nd and 3rd groupings. It seems one or both are typographical errors.

*- Sorry for the confusion. We replaced the tables with figures.*

---

## Referee Report (RR1)

General Comments

This revision addresses many of the previous comments but the manuscript would still benefit from further distinguishing between results within the GOCI domain and the full AHI domain, simplifying several of the figures to improve legibility, making use of additional tables to summarize statistics rather than relying on reading very small print in figures, and drawing additional conclusions. Specific suggestions are offered as follows.

Specific Comments

The addition of Table 2 greatly helps in distinguishing the sensitivity to the domain size, and a similar approach should be used throughout the manuscript to ensure apples-to-apples comparisons are being made. For example, as currently written, the analysis associated with figures 7 and 9 essentially says to ignore F3 and FM3 because they are for the larger domain. But instead, the authors could do the analysis in two groups: one in which data only within the GOCI domain are presented for all cases, and one which compares the large and small domain results for AMR, AES, F3, FM3 (which is already done now, in section 5.3 and Table 2). Consistent analysis of all results within the GOCI domain should allow the conclusions to be more easily shown. The effect of domain is a separate important facet, and again I really like Table 2 and its discussion.

Related, please clarify whether the results and analysis related to Figures 11 and 12 is conducted only within the GOCI domain or if it is mixing results with GOCI and AHI domains. I strongly feel it should be within the GOCI domain.

The postage-stamp graphics and small text in figures 7 and 9 make them nearly illegible. These figures could easily be replaced with simple tables that summarize all the statistics for all the cases (similar to Table 2) within the GOCI domain. I think the key points would clearly emerge, at least for the KORUS-AQ period: within the GOCI domain, all individual satellite products have similar statistics, the ensembles improve the statistics, and the ensemble-mean and MLE techniques appear to produce very similar results. With these new tables, I feel figures 7 and 9 could be omitted. Though if the authors feel there is some value in the figures, a representative subset could be shown, similar to how figures 4 and 5 have been recast to show AOD for only one example. I would suggest no more than 4 panels, to keep the figures legible (e.g., an AHI example, a GOCI example, F1, FM1). With these new tables, I also feel that Figure 13 can be omitted, as the tables would present the same information in a much more compact and easy to read form.

Similarly, figures 8 and 10 could be simplified by only showing representative examples. If the authors prefer to show all cases, I suggest that the 11-panel figures show only the GOCI domain in panels a, b, g, and k, and that a separate figure be used to show the AHI domain (perhaps with an outline of the GOCI domain) to elaborate on the low values in the extended domain.

I feel that Figure 6 and its discussion are out of place. I think it would make more sense for this analysis to come after the error analysis sections, because the estimated uncertainties of the geostationary satellite products have not been provided in the paper. (Maybe this uncertainty

information belongs in Section 2.) It is relevant here; it is not clear how meaningful it is to consider satellite AOD values in the 0.01 to 0.1 range.

Also on figure 6, the solid lines (left axis, dAOD) are not useful because the axis range is so large. dAOD rarely exceeds plus/minus 0.5 but the axis range is from -4 to 1. You probably need a separate plot, with appropriate axis range, to meaningfully show dAOD differences. Otherwise, the following statements about reduction of errors are not clearly demonstrated by the figure.

Section 6 should state high-level conclusions regarding the Remarks in Table 1. What can be said about the NRT vs all-available products, about the wider area of AHI, about the missing effect of GV1?

Technical Corrections

Line 168: Can you specify, what kind of data from JAXA?

Near line 232: Either here or in the Table 1 caption, add a statement that the 4 entries F1-F4 denote the ensemble-mean fusion technique and the 3 entries FM1-FM3 denote the MLE fusion technique.

Near line 256: Need to comment on the scatter apparent in Fig 1 for the squares (AOD>0.5). It is especially obvious at 2 and 5 UTC for NDVI>0.5, but also apparent as inconsistent temporal patterns between the NDVI bins. Also, there is a typo in the red label of Fig 1, it should be 0.5<NDVI.

Line 270 and following: Need to state the purpose of Figure 2, I don't understand what its significance is. Is it just a graphical illustration of how the bias is determined, or a means of assessing the degree to which the distributions are Gaussian? Also, the text isn't clear. The curves don't appear linear between plus/minus 1 to me. Though, what is the significance if they are not linear? Also, the multiple statements of excluding data beyond 2 SD seem repetitive.

Discussion of Figure 4: An additional statement should be made about Fig 4. The very small values shown everywhere in 4(f) show there is really very little difference between the two ensemble methods. There is only a fairly uniform small offset (bias) apparent.

Line 443: I think the word "smaller" should be "larger"

---

## Author Response (AR2)

**Reviewer 1:**

**Suggestions for revision or reasons for rejection (will be published if the paper is accepted for final publication)**

I reviewed the previous version of this paper. My main issue with the previous submission was that there were too many comparisons and similar unclear acronyms being used, which made it hard to find a coherent message. In this version the authors have reworked and streamlined the paper; I appreciate these efforts and think it makes some aspects easier to follow. However, I think some more work is needed. Some figures are still hard to understand (lots of nearly-overlapping similar dots, large numbers of panels which also tend to look similar). Some of the text is still unclear, out of order, or not needed (e.g. on line 166 there are 8 references for the simple statement that the AHI retrieval uses the Cox-Munk ocean surface reflectance; none of these references are to the Cox and Munk papers describing the actual equations). So it is still difficult to me to pick out a main message other than merging seems better than not merging. I think the relative merits of fusion over just doing the simple bias correction step should also be explored. From the figures, in general, it is difficult to pick out the key points and main message.

I therefore recommend some more revisions for clarity. I would be happy to review the revised version. I do not want to discourage the authors: this will be a good paper for the journal, and as I said in my previous review it is important that aerosol products from these geostationary sensors are given more attention. It is just not quite there yet, in terms of text and graphics.

→ Thank you for your valuable comments. We tried to reflect the comments by the reviewer, which improved the readability and contents of the manuscript.

Line 170: the use of 0.02 mg/m3 Chl still does not seem reasonable to me. I see the text has been changed from the previous submission, but "less contaminated" does not make sense.

[Figure]

Figure 1. Frequency distribution of JAXA CHL-a concentration in [mg m-3] for 1 May 0400UTC.

→ As JAXA CHL-a are retrieved for limited pixels with its own cloud masking, thus aerosol retrieval requires assumed CHL-a values for the remaining pixels without CHL-a products. Cloud masking is different for JAXA CHL-a and the AHI aerosol algorithm. Thus, $0.02$ mg/m$^3$ was assumed for the pixels without CHL-a data retrieved, to maintain the retrieval area of the AES product, which increased retrieval points by up to 2000 (as shown in the upper Figure). In the figure 1 of CHL-a frequency distribution, CHL-a concentration of $0.02$ mg/m$^3$ corresponds to a very small value in the frequency distribution. Thus it was described as 'less contaminated' in the viewpoint of aerosol retrieval. Table 1 shows the result of AOD calculated with different CHL-a values. With the original input of $0.913$ mg/m$^3$ CHL-a, AOD was retrieved to be 0.223. When $0.02$ mg/m$^3$ CHL-a was assumed as in this study, AOD retrieved was 0.225, which is very similar. AOD retrieved with $50.0$ mg/m$^3$ CHL-a was calculated as 0.222. Thus, the maximum difference in AODs due to the difference in CHL-a concentration is as small as 0.003.

Table 1 Sensitivity test for CHL-concentration of AES products under the following condition (SZA = 19.43 deg, VZA= 41.45 deg, RAA= 133.1 deg, and wind speed = 5.41 m/s).

| CHL-a concentration (mg/m$^3$) | Retrieved AOD at 550nm of AES algorithm |
| --- | --- |
| (original) 0.913 mg/m$^3$ | 0.223 |
| (test) 0.02 mg/m$^3$ | 0.225 |
| (test) 50.0 mg/m$^3$ | 0.222 |

According to Werdell and Bailey 2005, the in-situ CHL-a concentration using the High Performance Liquid Chromatography (HPLC) technique is 0.021-48.99 mg/m$^3$. Also, the NASA bio-Optical Marine Algorithm Data set's (NOMAD) minimum value for comparison with AHI CHL-a in Murakami 2016 paper appears around 0.02mg/m$^3$.

ref)

Werdell, P. J., & Bailey, S. W. (2005). An improved in-situ bio-optical data set for ocean color algorithm development and satellite data product validation. *Remote sensing of environment*, *98*(1), 122-140.

Murakami, H. (2016, May). Ocean color estimation by Himawari-8/AHI. In *Remote Sensing of the Oceans and Inland Waters: Techniques, Applications, and Challenges* (Vol. 9878, p. 987810). International Society for Optics and Photonics.

Lines 172-190: most of this text is a discussion of the relative merits of the MRM and ESR techniques. As such it probably belongs in the introduction, where these techniques are discussed, rather than in the algorithm description section.

→ Thank you for your comment; we moved these sentences to the introduction in lines 107-128 of the revised manuscript.

Lines 207-208: "the accuracy of GOCI, according to NDVI, has a negative bias for V1 and mostly a positive bias for V2" I don't understand what "according to NDVI" means here. Can this be reworded?

→ Thank you for your comments. This sentence was revised in lines 253-254 and as follows:

Line 263: The Sayer (2013) paper here in fact says the opposite of what the authors cite it for: it says that dAOD is not Gaussian, at least for Deep Blue data. See Figure 5(b) of that work. That is why the Deep Blue team have to define the expected error envelope and normalize to make it Gaussian. I think the authors are doing something similar here (in Figure 2) but the wording of the text implies the opposite.

→ Sorry for confusion. We removed the reference (line 335) and added Q-Q plot with the description in the manuscript in lines 344-349.

Figure 2: it would be good to add a line showing the theoretical QQ plot for a Gaussian distribution, to make it easier to show how close the data sets are. Also, it is not clear from the text/caption exactly what is plotted here. The text says "dAOD" but the caption says "z score". So is it dAOD divided by RMSE? This should be fixed.

→ Sorry for confusion. We added standard normal distribution (mean=0, std.=1) in the Q-Q plot. . The y-axis is normalized to std, and the text was revised. At both ends of the Q-Q plot, the sample quantile is more skewed than the theoretical value. However, it shows a symmetrical shape with respect to the point of 0.5 on the x-axis, so it still follows the Gaussian distribution.

Lines 275-282: I think this text is saying that, for each NDVI bin and hour of day bin (Figure 1), the mean bias is calculated and subtracted from the retrievals. Is that right? If so, can this paragraph be streamlined? If that is not right, can what is done be written more clearly?

→ Thank you for your advice. We revised these sentences in lines 354-356 of revised manuscript.

.

Section 4: this first part (page 9) uses the word "overestimate" a lot. However what the authors seem to mean is "this satellite combination is higher than that satellite combination". There is no discussion of the AERONET ground truth here so it is impossible to say whether one (or both) things being compared is overestimating or underestimating. It would be better to say that the two are "offset" relative to each other, as that only implies a difference, while "overestimate" implies an error.

→ Thank you for your comments.

We modified the word 'overestimate' to 'offset'. And we added validation result of respective satellite product with AERONET AOD summarized in table 2 (in Section 3.5) before Section 4(results section).

Figures 4, 5: these have 11 panels. Do we really need all these comparisons? Is there some better way to convey the intended message from these comparisons? It is hard to know exactly what details I am supposed to focus on here. I feel the figure is mixing both the comparisons of individual products with the results from individual merges. Maybe it would be best just to show campaign averages of the 4 baseline satellite products here so we can see the differences in them. Or show panels a-e (i.e. FM1 and the satellite products) in one figure and compare the other merges somehow. I am sorry I know it is difficult to present a large amount of information like this, and I am also not sure of the best way.

→ Thank you for your advice. Figures 4 and 5 were replaced with the difference of respective satellite product and average FM1 AOD to contrast the characteristics of each product. FM1 AOD is considered to be the representative fusion product as it includes.

Figure 6: this is interesting and to me shows that the MLE method helps at this site in low AODs. However it is hard to see some details because there are lots of near-overlapping dots from the measurements every 1 hour or so. And lots of big gaps from night time. So perhaps this plot could have the points plotted at daily scale instead? Or else add an additional figure which focuses on a short period (maybe a day or a few) so the x-axis is zoomed more effectively, and we can see how the products resolve the diurnal variability?

→ Thank you for your advice. The figure was revised as Fig 9 to the daily mean AOD during the KORUS- AQ period and corrected to show the diurnal variation of AOD from 11 May to 14 May.

Lines 345-357: I agree that the fused (especially MLE) products are better here. But it is not clear to me how much of this difference is due to the fusing, and how much is due to the bias correction step. Really we have two separate stages here: going from (1) satellite products to bias-corrected satellite products, and (2) from bias-corrected satellite products to the fused bias-corrected products. Unless I have misunderstood what is done here. There is obviously value in doing a bias correction (step 1), but the next step is less clear. Can the authors somehow separate this in the analysis?

→ The bottom figure shows the MLE results validation with the bias correction at the top and without applying bias correction at the bottom. Overall, RMSE, MAE, etc., are very similar. However, FM1 and FM2 of MBE slightly improve, but FM3 plays a role in worsening. However, the bias correction result for both %EE and %GCOS shows better accuracy in low AOD is improved.

Furthermore, the description of the bias correction effect was added to line 654-675 using FM3 and F3.

[Figure]

Figure 2. Validation of FM1-3 products with AERONET AOD during KORUS-AQ campaign (Top : Bias correction, bottom: no bias correction)

Lines 373-374: why compare to the MODIS DT EE when this is the expected uncertainty for a different algorithm and a different sensor? If the authors just want to provide a common reference uncertainty to benchmark against, ok, but then the paper has to be clear that this is not an "expected" error for any of the data sets (or merge) used here so is something of an arbitrary reference. Perhaps the GCOS goal (greater of 0.03 or 10% of AOD) would be a better comparison point since that is an international target not tied to one algorithm and sensor.

→ EE was used to compare common reference uncertainty to benchmark against MODIS as commented by the reviewer, and GCOS fraction was also added in lines 385-389. Also, we have specified 'EE' with respect to AERONET as EE for MODIS DT.

Section 5: it is a little strange to have the AERONET data and matchups described here when they were first used in Section 3.4 for the bias correction step. Some of this material should probably be moved earlier in the paper.

→ Thank you for your comments. We moved this part to the end of Section 3.3, because AERONET was also used for calculating RMSE.

Figures 7, 9: again, with 11 panels, it is hard to know what I should be looking for here. What is the main message of these figures? Are they necessary? Maybe it would be better if it were rearranged, with all "raw" satellite products on one row, all "averaged fused" on the middle, all "MLE" on the bottom? Or maybe it could be replaced with some plots of overall dAOD (combined to a smaller number of panels) and we hopefully see that the distribution of dAOD is narrower for the MLE results than the others? I am not sure but think somehow this should be streamlined. I am not sure we need to see 11 (mostly similar) scatter plots.

→ Thank you for your advice. We added a Table 2 and Table 3 to summarize the results and replace these figures.

Figures 8, 10: same comment about 11 panels and hard to know what is the main message I should be extracting here. I think the authors need to think about what they are trying to show here. If it is that MLE has a higher fraction in EE than others, then make a plot showing that more directly. There are a lot of colored dots and a lot of white space and nowhere to easily direct the reader to which panel(s) they should be comparing to tell whether something is better or worse.

→ The figure was revised as Fig. 6 the GCOS fraction of respective satellite product for each AERONET site, F1, and FM1 AODs.

Figures 11, 12: why is figure 11 shown as dots and horizontal envelope lines while 12 is dots with vertically drawn bars? I get that the lines and bars are conveying the same information but it would be better to present these two consistently (especially as the figure 12 caption says "as Figure 11"). I personally find the style of Figure 12 easier to see and get the main message (the bias corrected merged products are flatter with smaller errors). However it would be good to pick different colors for the top panels (raw data) compared to the other rows. This is because the eye will naturally compare the same color in each row but that is not relevant here because the colors of the satellite products in the top are not directly relating to the merges in the other rows. It makes sense to match colors between rows two and three because F1-F3 are equivalent to FM1-FM3, just not with the top row.

→ Thank you for your comment. We tried to add an error bar as requested by the reviewer, but what maintained the existing solid line due to poor visibility. Attached the picture indicated by the error bar below. It also improved

visibility by displaying colors consistently.

[Figure]

Section 6: if I understand correctly the authors recommend MLE over simple mean as the merging technique. This should probably be in the Abstract.

→ Thank you for your comments we added sentences in abstract (lines 32-34).

Section 6: In the Conclusions I would also appreciate some simpler, higher-level statements rather than repeating numbers about EE, etc mentioned a few pages earlier in the paper (no need to write them about twice). For example, about what the differences between the simple mean fusion experiments F1-F4 tell us, and between the individual MLE experiments FM1-FM3 tell us. For example the fact that FM1 is better than FM2 but F1 is worse than F2 is interesting. Comparing F1 with F2 (or FM1 with FM2) should tell us something about the quality of the NRT vs non-NRT data so it is interesting that the opposite results are obtained between simple mean and MLE approaches.

→ Thank you for your advice. We added these sentences in Section 6 (lines 753-765).

**Reviewer 2:**

**General Comments**

This revision addresses many of the previous comments but the manuscript would still benefit from further distinguishing between results within the GOCI domain and the full AHI domain, simplifying several of the figures to improve legibility, making use of additional tables to summarize statistics rather than relying on reading very small print in figures, and drawing additional conclusions. Specific suggestions are offered as follows.

→ Thank you for your valuable comments. We tried to reflect all comments by the reviewer, which improved the readability and contents of the manuscript.

**Specific Comments**

The addition of Table 2 greatly helps in distinguishing the sensitivity to the domain size, and a similar approach should be used throughout the manuscript to ensure apples-to-apples comparisons are being made. For example, as currently written, the analysis associated with figures 7 and 9 essentially says to ignore F3 and FM3 because they are for the larger domain. But instead, the authors could do the analysis in two groups: one in which data only within the GOCI domain are presented for all cases, and one which compares the large and small domain results for AMR, AES, F3, FM3 (which is already done now, in section 5.3 and Table 2). Consistent analysis of all results within the GOCI domain should allow the conclusions to be more easily shown. The effect of domain is a separate important facet, and again I really like Table 2 and its discussion. Related, please clarify whether the results and analysis related to Figures 11 and 12 is conducted only within the GOCI domain or if it is mixing results with GOCI and AHI domains. I strongly feel it should be within the GOCI domain.

→ Thank you for your suggestion. We revised all analyses to collocate the retrieved products for the GOCI domain and added the analysis for the AHI domain (excluding GOCI area) in section 5.3.

The postage-stamp graphics and small text in figures 7 and 9 make them nearly illegible. These figures could easily be replaced with simple tables that summarize all the statistics for all the cases (similar to Table 2) within the GOCI domain. I think the key points would clearly emerge, at least for the KORUS-AQ period: within the GOCI domain, all individual satellite products have similar statistics, the ensembles improve the statistics, and the ensemble-mean and MLE techniques appear to produce very similar results. With these new tables, I feel figures 7 and 9 could be omitted.

→ Thank you for your suggestion. We added a new table (Table 2 in revised manuscript) that has replaced these figures.

Though if the authors feel there is some value in the figures, a representative subset could be shown, similar to how figures 4 and 5 have been recast to show AOD for only one example. I would suggest no more than 4 panels, to keep the figures legible (e.g., an AHI example, a GOCI example, F1, FM1). With these new tables, I also feel that Figure 13 can be omitted, as the tables would present the same information in a much more compact and easy to read form. Similarly, figures 8 and 10 could be simplified by only showing representative examples. If the authors prefer to show all cases, I suggest that the 11-panel figures show only the GOCI domain in panels a, b, g,

and k, and that a separate figure be used to show the AHI domain (perhaps with an outline of the GOCI domain) to elaborate on the low values in the extended domain.

→ Thank you for your suggestion. The figure was revised by the GCOS fraction for respective satellite product, F1 and FM1 AODs at each AERONET site. We have added a statement to lines 505-513 along with a Figure 6 to mention improvement of the fused products' accuracy.

I feel that Figure 6 and its discussion are out of place. I think it would make more sense for this analysis to come after the error analysis sections, because the estimated uncertainties of the geostationary satellite products have not been provided in the paper. (Maybe this uncertainty information belongs in Section 2.) It is relevant here; it is not clear how meaningful it is to consider satellite AOD values in the 0.01 to 0.1 range. Also on figure 6, the solid lines (left axis, dAOD) are not useful because the axis range is so large. dAOD rarely exceeds plus/minus 0.5 but the axis range is from -4 to 1. You probably need a separate plot, with appropriate axis range, to meaningfully show dAOD differences. Otherwise, the following statements about reduction of errors are not clearly demonstrated by the figure.

→ Thank you for your advice. The figure was revised to the daily mean AOD during KORUS-AQ period and corrected to show the diurnal variation of hourly AOD (removed dAOD) for May 11-14. Also the figure 9 and discussion was moved after the error analysis section. The importance of low AOD is also mentioned earlier in Section 5.3.

Section 6 should state high-level conclusions regarding the Remarks in Table 1. What can be said about the NRT vs all-available products, about the wider area of AHI, about the missing effect of

→ Thank you for your comments. We added to discussion of Remarks (in Table 1) in Section 6 (lines 754-771).

**Technical Corrections**

Line 168: Can you specify, what kind of data from JAXA?

→ Thank you for your comment. We added the data type and URL.

Near line 232: Either here or in the Table 1 caption, add a statement that the 4 entries F1-F4 denote the ensemble-mean fusion technique and the 3 entries FM1-FM3 denote the MLE fusion technique.

→ Thank you for your comment. We added each entry information into the Table 1 caption.

Near line 256: Need to comment on the scatter apparent in Fig 1 for the squares (AOD>0.5). It is especially obvious at 2 and 5 UTC for NDVI>0.5, but also apparent as inconsistent temporal patterns between the NDVI bins. Also, there is a typo in the red label of Fig 1, it should be 0.5<NDVI.

→ Sorry for confusion. We revised Figure 1, also we added comments on the scattered results in lines 329-331.

Line 270 and following: Need to state the purpose of Figure 2, I don't understand what its significance is. Is it just a graphical illustration of how the bias is determined, or a means of assessing the degree to which the distributions are Gaussian? Also, the text isn't clear. The curves don't appear linear between plus/minus 1 to me. Though, what is the significance if they are not linear? Also, the multiple statements of excluding data beyond 2 SD seem repetitive.

→ Sorry for confusion. We addressed these issues in section 3.4. The bias was corrected using the Gaussian center value evaluated against AERONET for the evaluation period (Apr. 2018 to Mar. 2019 excluding EMeRGe period), which were applied for the KORUS-AQ and EMeRGe campaigns.

Discussion of Figure 4: An additional statement should be made about Fig 4. The very small values shown everywhere in 4(f) show there is really very little difference between the two ensemble methods. There is only a fairly uniform small offset (bias) apparent.

→ Thank you for your comments. Mean F1 and Mean FM1 appear similar because the RMSEs of AMR, AES, GV1, and GV2 used in this study are mostly similar to 0.1, and the weighting function used for MLE fusion becomes similar. However, Figure 4 has been revised, and Figure 4(f-k) has been removed.

Line 443: I think the word "smaller" should be "larger"

→ Thank you for your comment. However we removed Figure 13.

---

## Author Response (AR3)

I reviewed the previous versions of this paper. I appreciate the authors' continued work to improve clarity. This version of the paper is even more improved. However, a few more revisions are needed. These are quite minor; I would be happy to review again, but this might not be necessary, at the Editor's discretion. Additionally, the production office will need to do some copy-editing of the accepted manuscript prior to publication.

A) We appreciate deeply the reviewer's comments and suggestions again, which were very helpful in improving the final quality of our manuscript.

Line 141: aerosol height is not an AOP; this text should be changed. A similar comment applies to fine mode fraction (unless the authors are talking about optical depth fraction rather than mass or volume fraction, but the paper is not explicit).

A) Thank you for your comment. We revised the sentence in lines 140-143. Yes, we meant (optical) fine mode fraction in this manuscript.

Lines 302-311: Figure 2 still does not convince me that dAOD follows a Gaussian distribution. All the data sets show systematic deviations from the theoretical line in this figure. It is fine for the authors to say they are assuming it is Gaussian but then acknowledge that is a limited approximation. But don't say "it is Gaussian" pointing to the figure while the results clearly show it is not! This text needs to be corrected.

A) Thank you for your comment. We corrected the wording 'follow Gaussian distribution' to 'assume to follow the Gaussian distribution' in lines 315-316.'

Figure 3 and lines 424-425: is the y axis just the mean AOD bias for retrievals in that bin? This should be clearer; "Gaussian center" is not very obvious to me. If so I would just say "mean AOD bias" or similar in the figure and text.

A) Thank you for your suggestion. We corrected all the words 'Gaussian center' to 'mean AOD bias'.

Figure 8: caption says "Same as Figure8", I think this should be "Same as Figure 7".

A) Sorry for confusion. We corrected the 'Figure 8' to 'Figure 7' in the Figure 8 caption.